

Earth System
Dynamics

# Ubiquity of human-induced changes in climate variability

**Keith B. Rodgers**[1,2]**, Sun-Seon Lee**[1,2]**, Nan Rosenbloom**[3]**, Axel Timmermann**[1,2]**, Gokhan Danabasoglu**[3]**,
Clara Deser**[3]**, Jim Edwards**[3]**, Ji-Eun Kim**[1,2]**, Isla R. Simpson**[3]**, Karl Stein**[1,2]**, Malte F. Stuecker**[4]**,
Ryohei Yamaguchi**[1,2]**, Tamás Bódai**[1,2]**, Eui-Seok Chung**[5]**, Lei Huang**[1,2]**, Who M. Kim**[3]**,
Jean-François Lamarque**[3]**, Danica L. Lombardozzi**[3]**, William R. Wieder**[3,6]**, and Stephen G. Yeager**[3]

[1]Center for Climate Physics, Institute for Basic Science, Busan, South Korea
[2]Pusan National University, Busan, South Korea
[3]National Center for Atmospheric Research, Boulder, CO, USA
[4]Department of Oceanography and International Pacific Research Center, School of Ocean and Earth Science
and Technology, University of Hawai'i at Mānoa, Honolulu, HI, USA CE1
[5]Korea Polar Research Institute, Incheon, South Korea
[6]Institute of Arctic and Alpine Research, University of Colorado, Boulder, CO, USA

**Correspondence:** Keith B. Rodgers (krodgers@pusan.ac.kr) and Axel Timmermann (axel@ibsclimate.org)

**Abstract.** While climate change mitigation targets necessarily concern maximum mean state changes, understanding impacts and developing adaptation strategies will be largely contingent on how climate variability responds to increasing anthropogenic perturbations. Thus far Earth system modeling efforts have primarily focused on projected mean state changes and the sensitivity of specific modes of climate variability, such as the El Niño–Southern Oscillation. However, our knowledge of forced changes in the overall spectrum of climate variability and higher-order statistics is relatively limited. Here we present a new 100-member large ensemble of climate change projections conducted with the Community Earth System Model version 2 over 1850–2100 to examine the sensitivity of internal climate fluctuations to greenhouse warming. Our unprecedented simulations reveal that changes in variability, considered broadly in terms of probability distribution, amplitude, frequency, phasing, and patterns, are ubiquitous and span a wide range of physical and ecosystem variables across many spatial and temporal scales. Greenhouse warming in the model alters variance spectra of Earth system variables that are characterized by non-Gaussian probability distributions, such as rainfall, primary production, or fire occurrence. Our modeling results have important implications for climate adaptation efforts, resource management, seasonal predictions, and assessing potential stressors for terrestrial and marine ecosystems.

## 1 Introduction

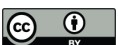 Faced with the prospect of substantial future climate change, mitigation and adaptation strategies are increasingly paramount. While mitigation efforts are concerned chiefly with limiting mean state changes, successful adaptation will also require understanding the potentially altered variability of the climate system (Sarachik, 2010). However, the way in which climate variability will change due to anthropogenic radiative forcing has not been extensively explored. The spectrum of observed regional-to-global climate fluctuations exhibits relatively sharp peaks and a broad noise background (Hasselmann, 1976; Franzke et al., 2020). Spectral peaks can emerge from a range of mechanisms, including astronomical forcings or internal climate instabilities such as for the El Niño–Southern Oscillation (ENSO). Moreover, these distinct features can be further influenced by climate processes acting on different timescales. Examples of non-linear "timescale interactions" are multiplicative (state-dependent) noise (Müller, 1987; Majda et al., 2009; Sardeshmukh and Sura, 2009; Sardeshmukh and Penland, 2015; Jin et al., 2007, 2020; Levine and Jin, 2010) and combination mode dynam-

ics (Stuecker et al., 2015b). How modes of variability will respond to greenhouse warming has been addressed in a number of previous modeling studies (Timmermann et al., 1999; Cai et al., 2018), albeit with conflicting results. In contrast, the sensitivity of the spectral background to human-induced climate change is less well-known. Identifying and characterizing human-induced changes in this spectral background, using for example Coupled Model Intercomparison Project (CMIP)-type coordinated modeling efforts, has proven difficult due to limited statistics.

The relatively recent advent of large ensemble simulations (henceforth termed large ensembles) conducted with Earth system models provides a new resource for addressing how climate and ecosystem statistics may evolve in response to anthropogenic forcing across a wide range of timescales (Deser et al., 2020; Schlunegger et al., 2020). Such large ensembles with global climate models have existed for more than 15 years (Zelle et al., 2005; Drijfhout et al., 2008; Branstator and Selten, 2009), but earlier studies expressed concern with aspects of process representation and therefore their results with regard to variability changes were inconclusive. Other studies have employed individual model simulations, small ($\leq 10$ members) ensembles, or CMIP multimodel ensembles (Rind et al., 1989; Raisanen, 2002; Huntingford et al., 2013; Screen, 2014; Stouffer and Wetherald, 2007; Wetherald, 2009) to address whether surface temperature and precipitation variability may change under global warming. To date large ensemble studies of changes in variance have mainly focused on specific quantities, timescales, or regions (Deser et al., 2020; Pendergrass et al., 2017; Maher et al., 2019, 2021; Haszpra et al., 2020). However, to our knowledge, the full power of the large ensemble framework has not been harvested to gauge broad-scale forced changes in climate statistics, including changes in variance, spectrum, patterns, and phase, for a wide range of quantities, regions, or timescales.

To study the sensitivity of higher-order climate statistics to anthropogenic climate change, we conducted a new 100-member ensemble of climate change simulations using the Community Earth System Model version 2 (CESM2) (Danabasoglu et al., 2020), which we refer to as CESM2-LE (Methods). The initialization and forcing are described in the Methods section and in Figs. S1–S5 in the Supplement. An ensemble of this size and duration with a CMIP6-generation Earth system model at 1° spatial resolution is unprecedented. A large number of improvements have occurred since CESM1-LE (Kay et al., 2015), as documented in the Methods section. In addition to improved parameterizations and process representation that increase model skill in representing a number of phenomena, a notable improvement is also seen in land processes within the Community Land Model Version 5 (CLM5).

CESM2-LE promises to provide an enhanced framework for documenting and understanding robust forced changes in internal variability, complementing our knowledge of mean-

state changes (Simpson et al., 2020; Fasullo, 2020). The simulations were performed for the 1850–2100 period with historical (1850–2014) and SSP3-7.0 (2015–2100) forcings. The choice to use 100 members was motivated by the challenges associated with identifying trends in higher statistical moments. A previous set of analyses performed with the Max Planck Institute Grand Ensemble (MPI-GE) (Milinski et al., 2020) explored the relationship between ensemble size and the accuracy of identifying forced changes in higher-order moments. Even taking into account differences in model architecture, and thereby model uncertainty in such estimates, their analysis with the MPI-GE nevertheless supports our decision to expand well beyond the 40 members chosen for CESM1-LE (Kay et al., 2015). To facilitate analysis over a broad range of timescales, substantial resources have been devoted to providing high-frequency output for the atmosphere, land, ocean, and cryosphere. Providing a clearer view of the patterns of altered climate variability should enable investigation of the mechanistic drivers of such changes and their implications for impacts of societal and ecosystem relevance. This study presents initial results on forced changes in internal variability across a range of fields and timescales in CESM2-LE, and it will serve as the reference publication for CESM2-LE.

## 2   Methods

### 2.1   Model configuration

The simulations consist of a 100-member large ensemble suite conducted with CESM2 with the Community Atmosphere Model version 6 (CAM6) (Danabasoglu et al., 2020), referred to here as CESM2-LE TS2. The simulations cover the period from 1850–2100 and follow the historical and SSP3-7.0 forcing protocols provided by CMIP6 (Eyring et al., 2016), although with some differences noted below for the representation of biomass burning in 50 of the 100 ensemble members. CESM2 has been demonstrated to fare well when evaluated against skill metrics with other models (Fasullo, 2020). The choice of the SSP3-7.0 scenario forcing follows CMIP6 recommendations (O'Neill et al., 2016) that emphasize the value of this relatively high forcing level precisely for the purpose of quantifying forced changes in natural variability. This choice should also provide a useful contribution towards an eventual CMIP6 large ensemble intercomparison.

The CESM2 components use nominal 1° horizontal resolution. Specifically, CAM6 has a resolution of 1.25° in longitude and 0.9° in latitude, and 32 vertical levels with a top at 2.26 hPa, or approximately 40 km. The ocean and sea ice models are the Parallel Ocean Program version 2 (POP2) (Danabasoglu et al., 2020; Smith et al., 2010) and the CICE Version 5.1.2 (CICE5) (Bailey et al., 2020). The nominal resolution of the ocean is 1° horizontally, with uniform spacing of 1.125° in the zonal direction and varying significantly in the meridional direction, with the finest resolution of $\sim 0.25°$

at the Equator. The ocean model provides 60 vertical levels, with 20 of these layers TS3 represented in the upper 200 m of the water column. CESM2 offers a number of improvements pertinent to our scientific interests relative to what was available for CESM1-LE (Kay et al., 2015). These improvements include advances in the surface boundary layer representation for the ocean (Li et al., 2016), as well as for cloud microphysics (Gettelman et al., 2015). The ocean biogeochemistry model used with the POP2 model is the Marine Biogeochemistry Library (MARBL), which represents an updated version of what was previously known as the biogeochemistry elemental cycle (BEC) (Moore et al., 2001, 2004, 2013; Long et al., 2021).

An important advance of great value to large ensemble investigations is achieved through new developments incorporated into CLM5 (Danabasoglu et al., 2020; Lawrence et al., 2019; Lombardozzi et al., 2020). The model addresses a number of well-known limitations relative to previous versions of CLM, including major improvements in simulated cumulative $CO_2$ uptake over the historical period (Bonan et al., 2019) and improved representation of the seasonal cycle of net ecosystem production (NEP) (Lawrence et al., 2019), which is highlighted in our analysis of projected forced phenology changes. Other notable features also included in CLM5 are the explicit representation of agricultural management and improvements in the implementation of the prognostic fire model (Lombardozzi et al., 2020; Li et al., 2013; Li and Lawrence, 2017). All CLM5 improvements found broadly across a range of simulated variables have been documented through evaluation of model simulations against the International Land Model Benchmarking (ILAMBv2.1) package and other analyses (Collier et al., 2018; Danabasoglu et al., 2020). We note that land model trajectories are sensitive to SSP scenarios that determine the spatial distribution and extent of land use and land cover changes (O'Neill et al., 2016).

As a more general complement to the research results considered in this study, we have also made available results from running the Climate Variability Diagnostics Package for large ensembles (CVDP-LE) (https://www.cesm.ucar.edu/working_groups/CVC/cvdp-le/, last access: 19 November 2021) (Phillips et al., 2020) for CESM2-LE, with graphical output available under https://climatedata.ibs.re.kr/data/cesm2-lens/lens-diagnostics (last access: 19 November 2021).

## 2.2 Large ensemble initialization

For the CESM2-LE initialization procedure, the experimental configuration was designed to respond to broad community demand for a mix of macro- and micro-perturbations (where for micro-perturbations members differ only in a small random perturbation applied at initialization). To satisfy this demand and allow for exploration of the impact of initialization type, it was decided to initialize members

from various years between 1001 and 1301 of a preindustrial simulation conducted with CESM2 (Danabasoglu et al., 2020). This was as far as the CESM2(CAM6) preindustrial simulation had reached at the time when the CESM2-LE project began, and by this point the top-of-the-atmosphere (TOA) global energy imbalance was relatively small and stable with a correspondingly small model drift (Danabasoglu et al., 2020). The years from the preindustrial control run used for initialization are highlighted in Figs. S1 and S3.

Micro-initializations start from four different years: 1231, 1251, 1281, and 1301. A total of 20 members were run for each start year, with ensemble spread introduced by a random perturbation to the atmospheric temperature field at initialization (through a CAM6 namelist variable referred to as "pertlim"), as was used for all members of CESM1-LE (Kay et al., 2015). Macro-initialization (one run for each initialization date) used initialization years {1001, 1011, 1021, …, 1191} by using 20 independent restart files at 10-year intervals over 1001–1191. It warrants mention for the case of the macro-perturbations that no explicit perturbation was required from the preindustrial control simulation. Taken together, if one includes one member from each of the micro-perturbation runs, then a total of 24 macro-perturbation runs are available.

Importantly, as can be seen in Fig. S1b, for the initialization points of years 1231, 1251, 1281, and 1301 were specifically chosen for the micro-initializations to correspond to years of maximum, decreasing, minimum, and increasing Atlantic Meridional Overturning Circulation (AMOC) transport, respectively, relative to the preindustrial control simulation. It is important to note that when using the large ensemble output, the initialization procedure should not be considered to produce members that are independent, or to have randomized modes of climate variability, for the years immediately subsequent to 1850. Considering the AMOC strength at 26.5° N as an example (Fig. S2), the ensemble mean AMOC strength for each of the micro-perturbation clusters initialized for years 1231, 1251, 1281, and 1301 of the preindustrial control run (averaged across 20 members for each case) converge only after several decades, indicative of the timescale over which the initial condition memory persists for AMOC. For this reason, our analysis with internal variability focuses on the period after 1960, more than a full century after initialization. Further quantitative exploration of the specific duration over which initial condition memory is retained is the subject of a separate ongoing study.

A generalized schematic for the initialization procedure is shown in Fig. S3, illustrating the organization of the simulations. The schematic also includes mention of the biomass burning emissions differences between two groups of 50 simulations, as described more fully in the next section. The macro-perturbation runs initialized at {1011, 1031, 1051, …, 1191} have greatly enhanced output at high frequency to meet the needs of broader community interests for large ensemble output. The temporally high-resolution output in-

cludes 6-hourly snapshots of three-dimensional temperature, winds, and specific humidity for the Coordinated Regional Climate Downscaling Experiment (https://cordex.org, last access: 19 November 2021) simulations, as well as output appropriate for the Cloud Feedback Model Intercomparison Project (CFMIP) Observation Simulation Package (COSP) (https://climatedataguide.ucar.edu/climate-data/cosp-cloud-feedback-model-intercomparison-project-cfmip-observation-simulator-package, last access: 19 November 2021).

## 2.3  Large ensemble forcing

A choice was made to use two different sets of forcing fields to represent the effects of variability in biomass burning emissions for CESM2-LE (see Figs. S4–S5). The biomass burning aerosol fluxes in CESM2 are imposed at the surface. As such, they are not prognostic, meaning that they are not generated by the model's internal prognostic fire model. The first 50 members of our large ensemble follow CMIP6 protocols (Van Marle et al., 2017), with biomass burning following the description in the CESM2 overview paper (Danabasoglu et al., 2020), and this forcing is referred to as BB_CMIP6. For the second set of 50 members, which we refer to as BB_CMIP6_SM (for smoothed biomass burning fluxes), the BB_CMIP6 biomass burning emissions of all relevant species for CAM6 were smoothed in time with an 11-year running mean filter. The averaging impacted variability in biomass burning fluxes over 1990–2020. Due to the inclusion of observations, the variability in biomass burning emissions during 1990–2020 is considerably stronger for BB_CMIP6 than the preceding and following periods. The smoothed forcing with BB_CMIP6_SM was designed to nearly conserve total emissions, while reducing the strong changes in interannual variability. The temporal smoothing of the forcing is applied to the biomass burning emissions at each grid point subsequent to being regridded to the CESM2 grid. The high 1990–2020 biomass burning variability case (ensemble members 1–50, or BB_CMIP6) relative to the smoothed forcing (ensemble members 51–100, or BB_CMIP6_SM) has a discernible impact on large-scale climate, as documented by the accelerated loss of September Arctic sea ice and northern hemispheric and tropical Pacific warming (Fig. S5a and c). Outside of the period 1990–2010, the impact of BB_CMIP6_SM relative to BB_CMIP6 for biomass burning emissions is not pronounced for simulated surface temperature, sea ice, or precipitation. It is for this reason that we selected the time intervals 1960–1989 and 2070–2099 for our analysis of variance changes in Figs. 2 and 4, for which the 100 ensemble members can realistically be considered to be part of the same population.

## 2.4  Minor corrections relative to previous versions

The code base for the BB_CMIP6_SM simulations (the second set of 50 members) incorporates corrections for two sets of errors that were present in the first set of 50 ensemble members (BB_CMIP6). The first pertains to the $SO_2$, $SO_4$, and gas-phase semi-volatile secondary organic aerosol (SOAG) emission datasets. For $SO_2$ and $SO_4$, the spatial patterns of the "shipping" and "agriculture+solvents+waste" components of forcing were inadvertently switched during the historical-to-projection transition, or more specifically at the start of 2015. The incorrect partitioning of $SO_2$ does not impact the results considered here, given that its components are summed before use. In contrast, the issue with $SO_4$ datasets can impact the model state evolution as the particle sizes and numbers differ for the $SO_4$ components. The SOAG emissions are calculated from several hydrocarbons, and they were not recalculated after an earlier bug correction in covering units of the lumped species for the biomass burning emissions. This issue was corrected, and diagnostics indicate that there are not any pronounced changes in the model solutions from these particular aerosol corrections.

The second correction introduced for the 50 BB_CMIP6_SM simulations concerns the presence of a sporadic large $CO_2$ uptake over land that was identified for the BB_CMIP6 runs. This large uptake is associated with a negative flux of carbon occurring at crop harvest time over a single time step. Although these large negative carbon flux component terms in autotrophic respiration are necessary for maintaining carbon balance, such $CO_2$ spikes are not realistic. To avoid these spikes, the associated $CO_2$ fluxes that occur over a single time step are distributed to the atmosphere over a timescale of approximately 6 months for the BB_CMIP6_SM simulations. Analysis indicates that these modifications for carbon between the BB_CMIP6 and BB_CMIP6_SM simulations did not result in any climate-changing impacts.

## 3  Results

### 3.1  Mean state changes

During the historical period the evolution of key simulated annual-mean climate indicators in CESM2-LE (Figs. 1 and S6) agrees well with observations. The range across the ensemble members, which results from internal variability and its forced changes, spans the observed climate state much of the time, with a notable exception being Southern Ocean sea ice (Fig. 1e). The results here and the general model behavior are qualitatively consistent with those of similarly forced CMIP6-generation models (Fasullo, 2020; Kwiatkowski et al., 2020; Arora et al., 2020), although projected temperature changes (Fig. 1c) are in the upper range of the CMIP6 models owing to the relatively high climate sensitivity of CESM2 (Gettelman et al., 2019). The progressive

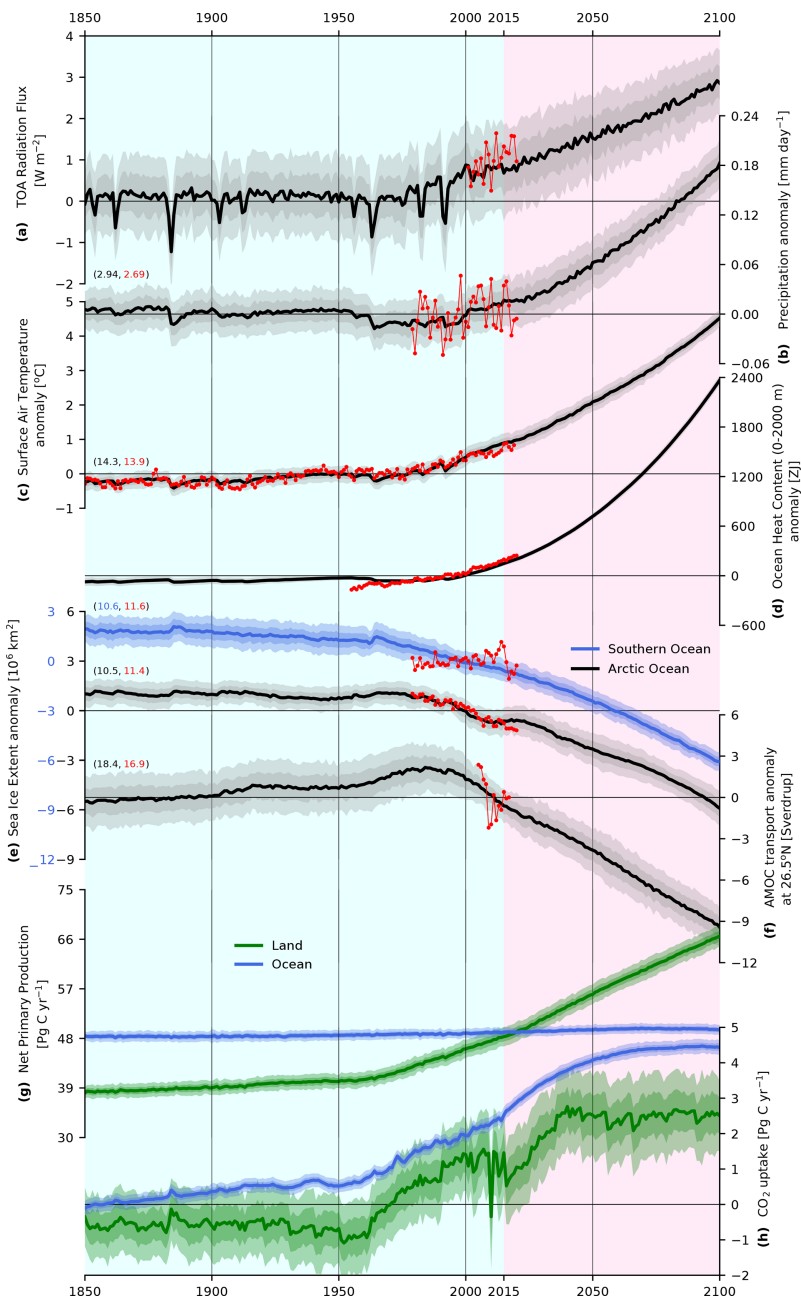

**Figure 1.** Annual mean evolution of global fields over 1850–2100 for 100 ensemble members. For model fields, bold lines represent ensemble means, and dark and light shading represent 1 standard deviation (SD) and 2 SD variability. Observational data are shown in red when appropriate. Portions of the figure with light-blue background shading indicate the historical period (1850–2014) while portions with light-red background shading indicate the projection period (2015–2100). **(a)** Top-of-atmosphere radiative imbalance (W m$^{-2}$) along with the CERES-EBAF product (Loeb et al., 2009, 2018); **(b)** anomalies of the global mean precipitation (mm d$^{-1}$) increasing 5.4 % between the 1850s and the 2090s, compared with the Global Precipitation Climatology Project (GPCP) (Adler et al., 2003, 2012); **(c)** anomalies of global mean surface temperature, increasing by 4.4 °C between the 1850s and 2090s, along with HadCRUT4 (Morice et al., 2012) anomalies over 1950–2019; **(d)** anomalies of ocean heat content integrated over the upper 2000 m, along with an observation-based product (Ishii et al., 2017); **(e)** anomalies of sea ice extent for the Arctic (black) and Southern Ocean (blue), with observed sea ice extent over 1979–2020 (Fetterer et al., 2017), and with the vertical scales of the anomaly plots offset to facilitate comparison; **(f)** Atlantic Meridional Overturning Circulation (AMOC) transport anomalies at 26.5° N, with RAPID array observations (Frajka-Williams et al., 2019); **(g)** globally integrated net primary productivity (NPP) over the ocean (blue; increase of 2.7 % between the 1850s and 2090s), and over land (green); and **(h)** globally integrated net CO$_2$ fluxes over the ocean (solid blue) and integrated net CO$_2$ flux (net biome production, or NBP, including fire and land-use change) over land (green) with all quantities in **(g)** and **(h)** in units of Pg C yr$^{-1}$. For each case, where observational products are included, anomalies are calculated with respect to the period spanned by the observations. For anomaly fields, printed numbers represent the absolute mean of the ensemble mean of CESM2-LE (black or blue numbers) and the observational product (red numbers).

weakening of the AMOC at 26.5° N in CESM2 over the 21st century (Fig. 1f) is largely consistent with other CMIP6 models (Weijer et al., 2020). We also find a substantial increase in land primary productivity (Fig. 1g), which contributes to the uptake of carbon in the terrestrial biosphere. Marine net primary productivity (NPP) (Fig. 1h) remains relatively constant throughout the simulation, and the overall uptake of carbon by the ocean reflects the re-emergence of anthropogenic carbon into the mixed layer (Toyama et al., 2017; Rodgers et al., 2020) and changes in the $CO_2$ buffering capacity of seawater (Revelle and Suess, 1957). For the analysis that is presented in Fig. 1 for sea ice, daily-mean output fields are used for both the model and the data TS4 product. In representing sea ice extent a threshold of 15 % was used, whereby a grid cell is identified as being ice covered if it has a concentration of sea ice above 15 %. For the net land fluxes of $CO_2$, we use the variable net biome production which includes the effects of not only photosynthesis and respiration, but also fire and land-use change.

The pattern of mean state surface temperature change, shown as the difference between the periods 2070–2099 and 1960–1989 (Fig. 2, central panel; 2 m reference temperature shown in Fig. S6), exhibits preferential warming of the eastern relative to the western equatorial Pacific, Arctic amplification, and a pronounced warming hole over the subpolar North Atlantic. These features are associated with the known mechanisms of the enhanced equatorial warming pattern (Xie et al., 2010), more positive polar feedbacks (Goosse et al., 2018) including the Arctic heat capacitor (Chung et al., 2021), and the slowdown of the AMOC (Rahmstorf et al., 2015; Menary and Wood, 2018), respectively. For precipitation (Fig. 2, central panel; Fig. S6e), changes include marked precipitation increases along the equatorial Pacific, within the Arctic Ocean and decreases over the subtropical regions (Stocker et al., 2013).

## 3.2 Forced changes in amplitude, frequency, and phase

Figure 2 illustrates the ensemble-aggregated TS5 Fourier amplitude spectra and probability density functions (PDFs) for five key climate and ecosystem quantities (complementary quantities are shown in Fig. S7). The choice of variables reflects an interest in both climate and ecosystem dynamics, as well as societal relevance in terms of adaptation and resource management. The decision to represent Fourier amplitude spectra was motivated by our desire to enrich our understanding of the amplitude of perturbations across different timescales. For the spectral analysis in Fig. 2, each fast Fourier transform (FFT) spectrum is calculated for the time series of raw data over a given variable for the full 30-year interval. This includes all timescales shorter than 30 years and longer than 2 days (months) for daily (monthly) time-resolution data. The spectrum is calculated first at each horizontal grid point and for each ensemble member and then averaged over the designated region and over the 100 en-

semble members. Due to the relatively large degree of aggregation for each field, it was not necessary to apply windowing to avoid spectral leakage. The surface chlorophyll concentration fields analyzed here represent total chlorophyll concentrations taken as a sum of diatom, diazatroph, and small phytoplankton chlorophyll concentrations. The AMOC in Fig. S7 is defined as a maximum transport at 26.5° N. For the spectrum of internal variability of the AMOC, the ensemble-mean is subtracted from the raw data to remove the forced response

For a wide range of Earth system variables, we find substantial changes of the projected 21st century probability distributions, impacting mean state, variance, and higher-order statistical moments (Fig. 2). Human-induced alterations of climate spectrum and probability distribution could translate into changes in the average return time of climate and extreme events. Averaging the spectra over 100 ensemble members and individual grid boxes within each region creates enough data to reveal spectral characteristics that might otherwise be obscured. The dominant feature for most quantities examined is the spectral peak at the annual frequency, along with higher-order harmonics that result from deviations of the seasonal cycle from a pure sinusoid. Future changes of the annual cycle overtone spectrum can be caused by forced non-sinusoidal distortions of the annual cycle, generated for example by shifts in phenology, as discussed below. For nearly all variables under consideration, the seasonal cycle amplitude responds to the external forcing. Near-annual combination modes (C modes CE2) of ENSO and the seasonal cycle (Stuecker et al., 2015a) and its overtones can be clearly identified in some spectra, particularly for precipitation over the equatorial Pacific. In addition to representing the C modes as deterministic components of the system, CESM2-LE also exhibits shifts in the frequency of the C modes due to future reductions in ENSO's dominant frequency (Fig. 3a). The C-mode peaks also strengthen in the future, reflecting that the amplitude of precipitation and the corresponding C-mode-generating non-linearity increase at both ENSO and annual frequencies.

For most of the variables shown in Fig. 2 (and Fig. S7) there are changes in the amplitude of the spectrum across the entire range of frequencies from synoptic to intra-seasonal to interannual to decadal, revealing the ubiquity of variance changes. Importantly, frequency-independent shifts in variance can be seen in the three variables shown here, which exhibit a strong non-Gaussian skewed PDF, namely the spectra of California wildfire occurrence, surface chlorophyll concentrations over the subpolar North Atlantic (40°–60° N, 60°–15° W), and precipitation over the Niño 3.4 region (5° S–5° N, 170°–120° W). For these positive variables with their highly skewed probability distributions, forced changes in the mean state are accompanied by a stretching (squeezing) of the associated PDFs, thereby causing enhancement (or reduction) of variance and extremes. Changes of this type have previously been considered for more specialized cases

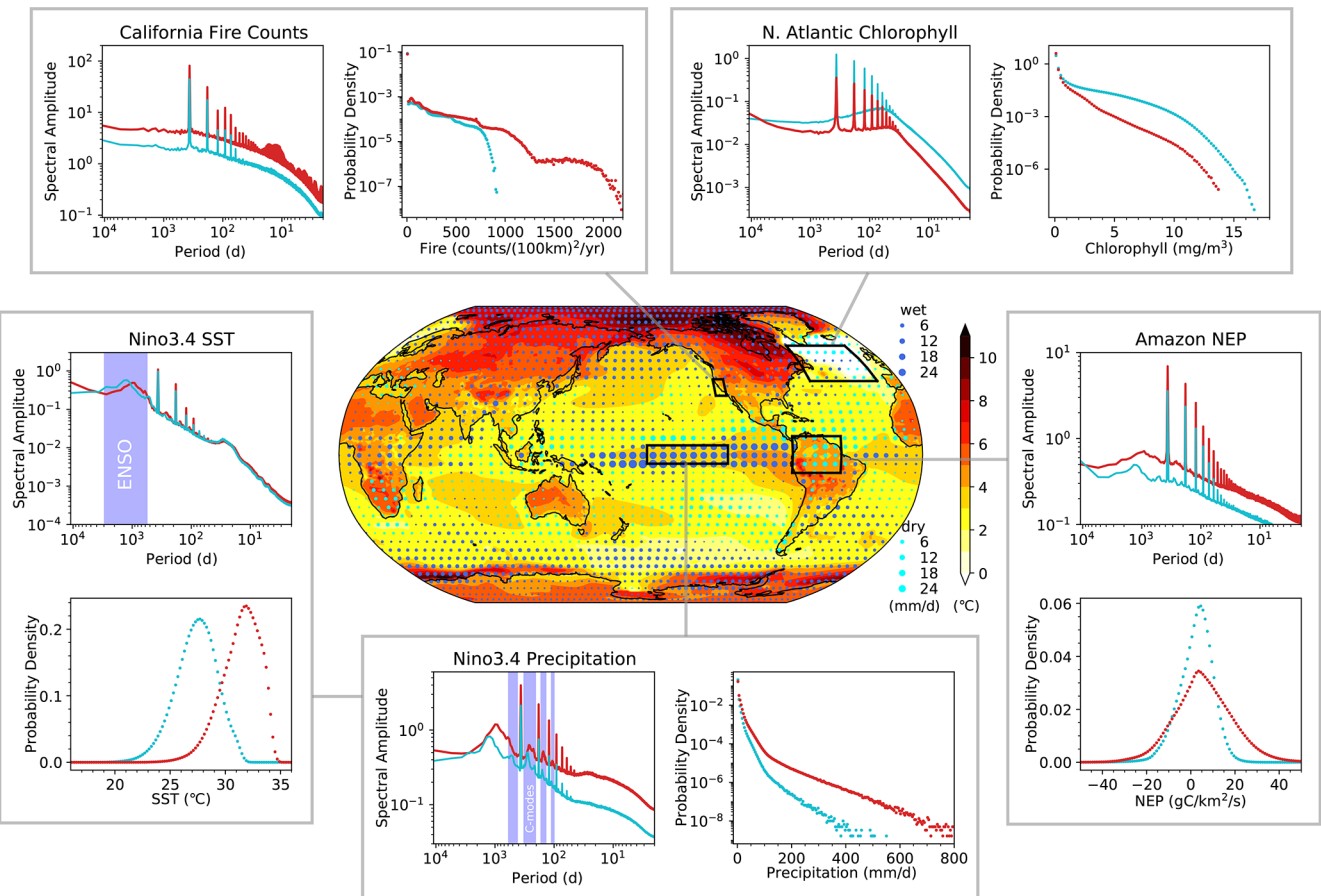

**Figure 2.** Changes in the Fourier amplitude spectrum of historical (1960–1989) to future (2070–2099) climate variability in CESM2-LE. The center map shows historical-to-future changes in surface temperature (shaded, °C) and precipitation (solid blue/cyan dots, mm d$^{-1}$). Each pairing of panels shows historical (cyan) and future (red) spectra and PDFs for five different variables over four different regions. The spectra are considered over the respective periods, 1960–1989 (historical) and 2070–2099 (future), thereby including the trend, and PDFs are considered for all days over 1980–1989 and 2090–2099 to minimize the impact of the trend. From upper-left clockwise, each pair of panels shows fire occurrences in California (32°–41° N, 125°–118° W, land only), surface chlorophyll concentrations in the North Atlantic subpolar gyre (40°–60° N, 60°–15° W), net ecosystem production (NEP) in the Amazon (10° S–10° N, 80°–50° W, land only), precipitation over the Niño 3.4 regions (5° S–5° N, 170°–120° W), and sea surface temperature (SST) over the Niño 3.4 region. The spectra are calculated for daily time series at individual grid points including both forced responses and internal variability and using 30-year intervals. Subsequently the spectra are averaged over the grid points in each region. Sharp spectral peaks are associated with the annual cycle and its non-sinusoidal components, which generate high-order harmonics. Shaded areas for spectra of precipitation and temperature in the Niño 3.4 region correspond to the timescales of the El Niño–Southern Oscillation (ENSO) and ENSO-annual-cycle combination modes (Stuecker et al., 2013) (C modes). Spectra are shown as amplitude, with the units being the same as the $x$ axes for the PDFs. PDFs of positive variables (California fire counts, North Atlantic surface chlorophyll, and Niño 3.4 precipitation) are shown with logarithmic $y$ axes. The fields in the center panel are presented in more detail in Fig. S6, except that there 2 m reference temperature is used rather than surface temperature. A suite of complementary spectral and PDF analyses to those shown here are presented in Fig. S7.

using the Wasserstein distance (Ghil, 2015; Robin et al., 2017; Vissio et al., 2020). For white noise processes, the associated variance changes manifest as timescale-independent variance changes, thereby accounting for the shown spectral background shifts. For California fire counts and Niño 3.4 precipitation, mean state increases are therefore also accompanied by increases in variance occurring over a wide range of timescales. For North Atlantic chlorophyll concentrations, the mean state decrease is associated with a timescale-independent decrease in variance, with expected impacts for higher trophic levels in the ocean, leading to potential disruptions to ecosystems.

For variables that are less skewed, a diversity of responses is found. Forced changes in sea surface temperature (SST) variability in the Niño 3.4 region are confined to interannual timescales in association with a decrease in ENSO amplitude and a slight shift toward higher frequencies. On the other hand, for NEP over the Amazon, reflecting natural $CO_2$ ex-

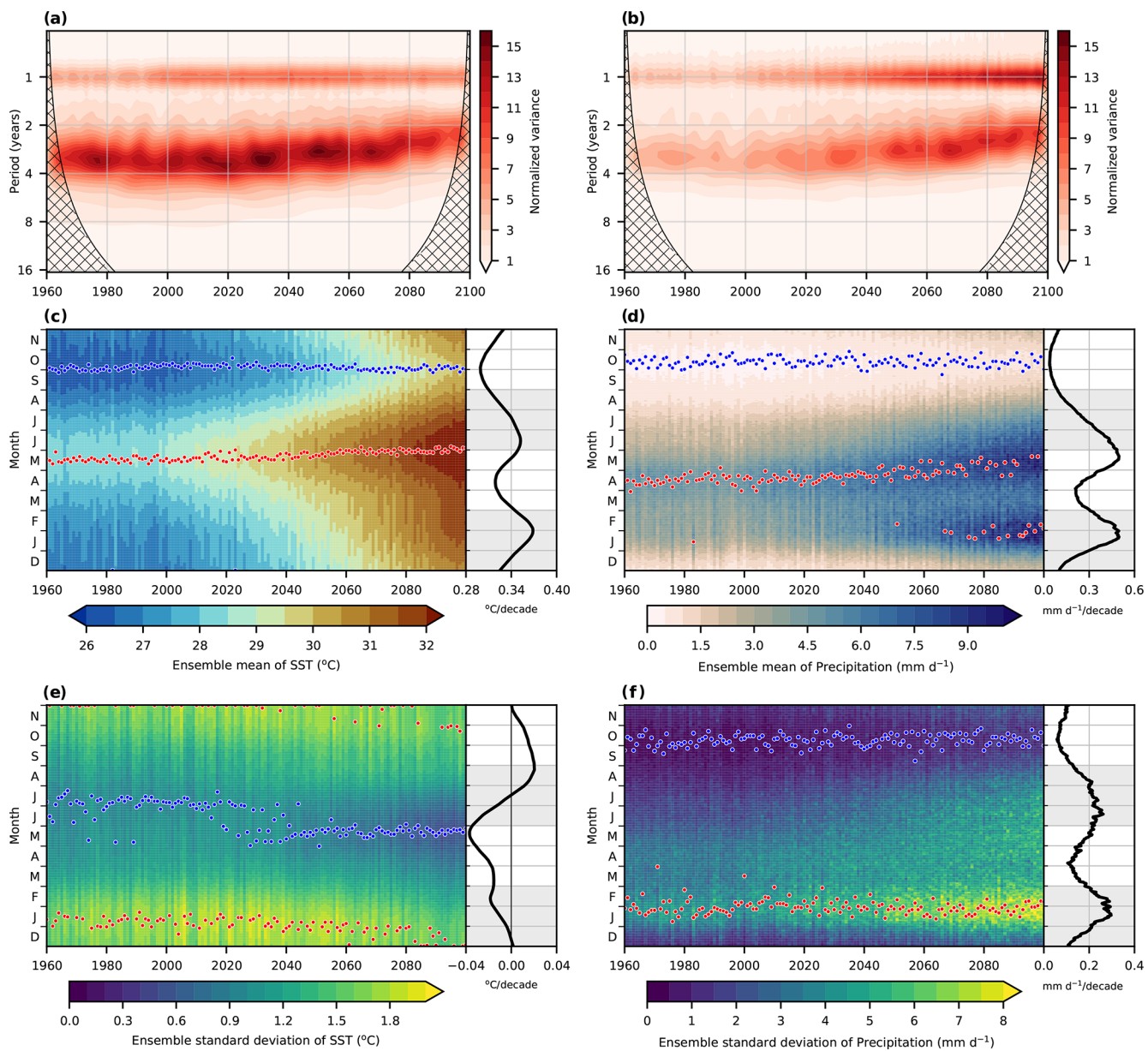

**Figure 3.** Changes in the dominant frequencies and seasonal variance of sea surface temperature (SST, left) and precipitation (right), in the Niño 3.4 region (5° S–5° N, 170°–120° W). The top row shows the wavelet power spectra of Niño 3.4 **(a)** SST and **(b)** precipitation using a Morlet wavelet, normalized by $\overline{\sigma}^{-2}$, where $\overline{\sigma}$ is the ensemble mean standard deviation of the respective Niño 3.4 time series (Torrence and Compo, 1998). The $y$ axis shows TS6 the equivalent Fourier period in years. The hatching indicates regions where the wavelet spectrum is not trustworthy due to edge effects. Prior to calculating the wavelet spectra, the time series were detrended by subtracting the ensemble-mean annual means, which were linearly interpolated to a monthly time step. The middle row shows the ensemble-mean of Niño 3.4 **(c)** SST and **(d)** precipitation indicated for each day (ordinate) and year (abscissa) using daily output. The red (blue) dots indicate the maximum (minimum) daily values of each year. The black line to the right in panels **(c–f)** indicates the linear trend over 1960–2100. The bottom row shows the same as for **(c)** and **(d)**, but for the across-ensemble SDs of **(e)** SST and **(f)** precipitation.

change between the land and the atmosphere, there is an increase in variance over all timescales, accompanied by a shift in the broad interannual peak towards higher frequencies.

To demonstrate the significance TS7 for the spectra considered in Figs. 2 and S7, an example is given in Fig. S8 for pre-

cipitation over the Niño 3.4 region for the same 1960–1989 (blue) and 2070–2099 (red) time intervals. The grey shading indicates the 95 % confidence interval (1.96 × standard error). For each ensemble member, we first spatially averaged the spectra at individual grid points over the Niño 3.4

region and then calculated the ensemble-wise (or across-ensemble) [TS8] error using the 100 spectra for the full 100 ensemble members. This approach appropriately handles spatially correlated data in the calculation of the confidence intervals and is known as the block bootstrap method, where the block means are the spatial means. The estimated confidence intervals in Fig. S8 indicate that the spectra for 1960–1989 and 2070–2099 are statistically different. It is worth noting here that if samples at different grid points are treated as being independent samples, the confidence intervals become much narrower, in which case the two spectra in Fig. S8 are even more statistically significant in their difference [TS9].

We next turn our attention to an expanded view of the temporal evolution of both frequency and amplitude modulations of SST and precipitation over the Niño 3.4 region over the period 1960–2100. For the wavelet analysis (Torrence and Compo, 1998) in Fig. 3, we apply a Morlet wavelet normalized by $\overline{\sigma}^{-2}$, where $\overline{\sigma}$ is the ensemble mean standard deviation of the respective time series. For analyses of patterns of changes in variance, an adjusted Welch's $t$ test (Torrence and Compo, 1998) was applied. [TS10] The general approach is to first calculate the equivalent sample size $\hat{n}$, to account for potential serial correlations of the time series. This is then used to calculate the degrees of freedom for the Welch's $t$ test, which is an adjusted version of the Student's $t$ test that allows for the two samples to have unequal variance (i.e., heteroskedasticity). First, the decorrelation timescale $T_e$ was calculated at each grid point, and for each period, based on the $e$-folding timescale of the autocorrelation function $r(\tau)$, defined as the smallest lag $\tau$ for which $r(\tau) < e^{-1}$. Then the equivalent sample size $\hat{n}$ was defined as $\hat{n} = N/T_e$, where $N = 30$ is the total sample size in our case. The equivalent sample size was then used to calculate the degrees of freedom of the standard Welch's $t$ test. Note that this test may still be liberal if the equivalent sample sizes are small, i.e., in areas of high serial correlation.

Ensemble wavelet analysis of SST (Fig. 3a) and precipitation (Fig. 3b) within the Niño 3.4 region has been conducted after first removing the ensemble-mean trend over the full period from each ensemble member, while retaining the seasonal cycle. The wavelet analysis is conducted for each ensemble member and then averaged. Our motivation for retaining the seasonal cycle stems from an interest in illustrating timescale interactions between ENSO and the seasonal cycle with the full power of large ensemble statistics. The annual cycle and ENSO interact with each other in a complex way, with the annual cycle itself being a forced coupled air–sea mode (Xie, 1994). This interaction gives rise to combination modes (Stuecker et al., 2015b), frequency entrainment (Timmermann et al., 2007), and ENSO's phase-locking and seasonal variance modulations (Stein et al., 2010, 2014). Not only does the annual cycle in the equatorial Pacific influence the amplitude and phase of ENSO, but ENSO also impacts the seasonal cycle.

We consider the normalized variance to highlight the amplification above the white noise level, and in contrast to Fig. 2 represent variance with a linear scale to emphasize temporal modulation of the amplitude of the maxima. For SST a clear separation is seen between the maxima for interannual variability and the annual cycle (Fig. 3a). At interannual timescales, there are two notable features. The first is a shift in the ENSO peak period from 3.5 to 2.5 years between the end of the 20th century and the end of the 21st century. The second feature with interannual variability is that variance does not change monotonically but rather exhibits a maximum midway through the 21st century, similar to what has been reported elsewhere (Kim et al., 2014). This stands in contrast to precipitation over the same region (Fig. 3b), for which there is a monotonic increase in variance, following a similar shift in the period of the peak that was found for SST. For precipitation, the amplitude of the seasonal cycle increases over 1960–2100, consistent with the notion of variability enhancement over the tropics due to thermodynamic and dynamic processes (Yun et al., 2021).

The forced changes over 1960–2100 in the structure of the seasonal cycle for the ensemble mean of SST (Fig. 3c) and precipitation (Fig. 3d), as well as the across-ensemble standard deviation of SST (Fig. 3e) and of precipitation (Fig. 3f) are also considered for the Niño 3.4 region using daily-mean model output. The maximum (red dots) of ensemble-mean SST occurs in May and the minimum (blue dots) in October in the late 20th century (Fig. 3c), with both showing monotonic increases over 1960–2100. The maximum shifts to 2 weeks later and the minimum shifts to 2 weeks earlier by the end of the 21st century, with this modest perturbation to the phase of the seasonal cycle being accompanied by a modulation of seasonal amplitude. The ensemble-mean seasonal amplitude in precipitation (Fig. 3d) occurs approximately 1 month before the ensemble-mean maximum in SST (Fig. 3c), and a second maximum in precipitation in late January becomes evident during the second half of the 21st century. On the other hand, the ensemble-mean minimum in precipitation occurs approximately 2 weeks after the local minimum in temperature. The increase in the amplitude of the seasonal cycle is thereby accompanied by changes in the phasing of the seasonal cycle for both SST and precipitation.

The mechanisms responsible for the phasing of maximum precipitation leading maximum temperature over the Niño 3.4 region over seasonal timescales (red dots in Fig. 3c and d) have been considered previously in published literature (Xie, 1996; Xie et al., 2010; Williams and Patricola, 2018; Stuecker et al., 2020). Current understanding maintains that seasonal precipitation phasing is largely driven by meridional SST gradients and is thereby not directly tied to the phasing of seasonal SST variations in the Niño 3.4 region. In other words, the phase relationship between precipitation and SST is not surprising, as moisture convergence is in part determined by non-local SST conditions.

The seasonally stratified maximum across-ensemble SD in SST (Fig. 3e), associated with peak ENSO variability, exhibits a trend towards an earlier occurrence by approximately 1 month over 1960–2070. This is accompanied by a modest decrease in amplitude (line plot). The across-ensemble SD minimum for SST occurs in July for the 20th century, with a secondary minimum in the across-ensemble SD developing over the first half of the 21st century in May. Subsequently the across-ensemble SD minimum in May becomes more pronounced and becomes the dominant minimum in the across-ensemble SD of SST by the end of the 21st century. For the across-ensemble SD of precipitation (Fig. 3f), there is a monotonic strengthening of the seasonal maximum in late January, corresponding roughly to the time of peak ENSO variability, and a weakening of the seasonal minimum in October, over the interval 1960–2100. Whereas the seasonal minimum in the across-ensemble SD of precipitation (Fig. 3f) occurs nearly in phase with the seasonal minimum of ensemble-mean SST (Fig. 3c), the seasonal maximum for the across-ensemble SD of precipitation does not coincide with the seasonal maximum of ensemble-mean SST. Rather, it coincides with the secondary seasonal maximum in ensemble-mean precipitation in late January (Fig. 3d).

## 3.3 Changes in variance and co-variance patterns

Along with modulations in the frequency domain, the spatial patterns of variance are altered in response to changing climate conditions. The analysis of patterns of variance and co-variance in Fig. 4 uses across-ensemble calculations of annual-mean ensemble SDs TS11. These calculations entail first calculating the SD TS12 across all ensemble members for the same time record. Subsequently averaging is done across time. This sequence TS13 was chosen to avoid spurious amplification of variability due to the non-trivial forced variations in precipitation and surface temperature driven by volcanic aerosols over the historical period. TS14 For the case of surface, averaged over December–January–February (DJF) (Fig. 4a) and precipitation for DJF TS15 (Fig. 4b), the across-ensemble SDs were first calculated separately over all years spanning 1960–1989 and 2070–2099, and then averaged over the two respective periods. The intention with the calculation of both across-ensemble SDs and correlations is to harness the full power of the large ensemble and is analogous to the empirical orthogonal function (EOF) EOF-E snapshot method (Maher et al., 2018).

We begin by considering interannual TS19 variance changes in boreal winter (DJF) by evaluating relative changes in the across-ensemble SD of surface temperature and precipitation for the same periods as with the spectra in Fig. 2 (1960–1989 and 2070–2099). The background across-ensemble SD averaged over 1960–1989 is shown in shading (Fig. 4a and b) TS20. Surface temperature (Fig. 4a) reveals modest decreases in variability across the equatorial Pacific and Indian oceans, consistent with Fig. 2. Variability decreases over

much of the higher latitudes of the Northern Hemisphere (Screen, 2014; Screen et al., 2015; Holmes et al., 2016; Sun et al., 2015; Schneider et al., 2015), with exceptions over the Arctic and the North Atlantic, and with exceptions in the Southern Hemisphere found over southern Africa and parts of Antarctica (Fig. 4a). For precipitation (Fig. 4b) a relative increase in SD is seen over most regions with particularly pronounced enhancements occurring in the eastern equatorial Pacific, the Indo-Pacific warm pool including the South Pacific Convergence Zone, the western Arabian Sea, the poles, and most land areas. The equatorial Pacific changes represent an eastward broadening in the centers of convection in response to the enhanced equatorial Pacific warming and the reduction of the overall zonal SST gradient (Fig. 2, center). In contrast, there is a decrease in the northern equatorial Atlantic Ocean as well as in some trade wind regions of the eastern Pacific.

Another important question to address is whether greenhouse warming can also impact the co-variability of different climate components and the global teleconnections of major modes of climate variability. This is illustrated here by examining the projected changes in the local correlation coefficients between the Niño 3.4 SST index and surface temperature from 1960–1989 and 2070–2099 (Fig. 4c), with the background correlation coefficients shown in shading and their respective future changes shown in circles. Our analysis reveals a systematic strengthening of ENSO's remote temperature correlation over the Amazon basin and in the equatorial Atlantic, the Philippines and Japan in the western Pacific, throughout Africa, in northern India and across eastern Canada and the southern US. Co-variance decreases over western Canada and Alaska, and zonally across the equatorial Indian Ocean.

The future changes in the correlation between the Niño 3.4 index and precipitation (Fig. 4d) indicate a pattern of enhanced co-variance over the western Pacific region surrounding the Philippines, much of Africa and South America, and western China, as documented by the background correlation coefficients and their future changes having the same sign. In other words, in these regions we see stronger ENSO teleconnections under future global warming, which in turn could translate to increased predictability of climate in the regions on seasonal to interannual timescales, but also stronger impacts. In contrast, decreased precipitation co-variance with ENSO is found for North America over the Pacific Northwest as well as much of the southern US and Mexico, as well as over Columbia and Venezuela, Bangladesh and Myanmar, parts of eastern Australia, and parts of eastern Siberia. Taken together, the global pattern of ENSO precipitation co-variance changes (Fig. 4d) is due to a combination of simulated weakening of ENSO SST variability (Fig. 4a) and eastward expansion of the region of maximum convective activity in the equatorial Pacific (Fig. 4b) (analysis for the June–July–August (JJA) season in shown in Fig. S9), and likely other projected changes of the background atmo-

Please note the remarks at the end of the manuscript.

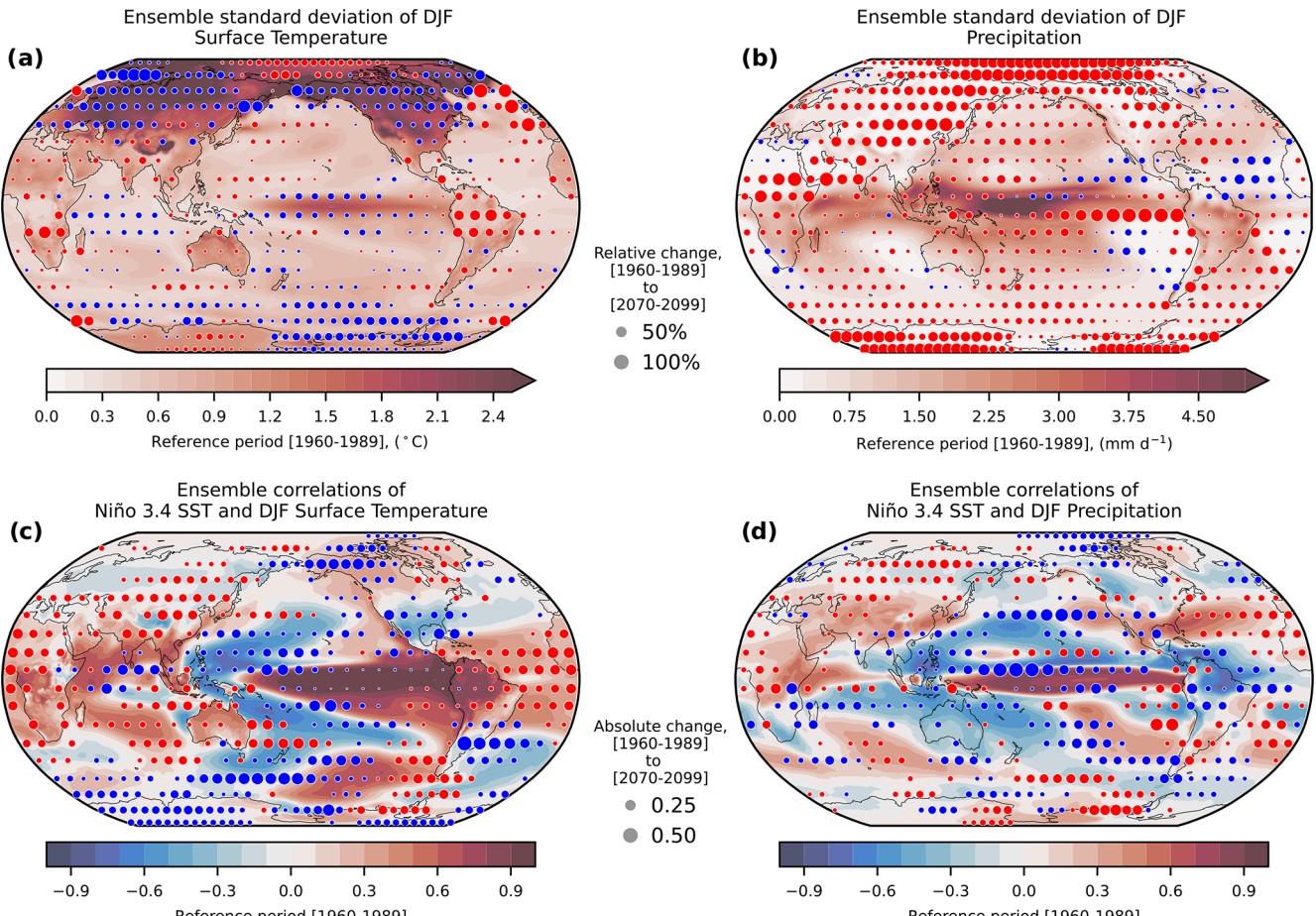

**Figure 4.** Changes in the patterns of interannual variability and Niño 3.4 correlation coefficients of December–January–February (DJF) surface temperature and precipitation. In the top row the color shading shows the time-averaged absolute TS16 across-ensemble SD of the DJF seasonal mean surface temperature **(a)** and precipitation **(b)** for the period 1960–1989. Circles show the relative changes in the SD between 2070–2099 and 1960–1989, where insignificant change ($p \geq 0.05$) has been removed. Statistical significance of the changes (circles) was determined based on the $p$ values of the two-sample Welch's $t$ tests for the equality of temporal means of the SDs, with the equivalent sample sizes adjusted to account for serial correlations (Sect. 3.3). In the bottom row the color shading shows ensemble-wise TS17 correlations of the Niño 3.4 index with surface temperature **(c)** and precipitation **(d)** anomalies for DJF, averaged over the period 1960–1989. Circles show the absolute change in correlations between 2070–2099 and 1960–1989, where statistically insignificant changes ($p \geq 0.05$) have been removed. The Niño 3.4 index for ENSO is the spatial average of sea surface temperature within 5° S–5° N, 170°–120° W. Statistical significance of the changes (circles) was determined based on the $p$ values of two-sample Student's $t$ test of the Fisher $z$-transformed correlation coefficients (Timmermann et al., 2014). Note that the $t$ test treats the ensemble standard deviations TS18 and correlations as stationary and serially uncorrelated within either of the two periods. For all four panels, the circles represent subsampled fields at 10° intervals over the global domain. The corresponding analysis for June–July–August (JJA) is presented in Fig. S9.

spheric circulation. There are a number of outstanding challenges in interpreting mechanistically how ENSO teleconnections change in response to anthropogenic forcing, including the relative role of local diabatic forcing and modulations of ENSO (Taschetto et al., 2020). We anticipate that the large ensemble analyses here will complement efforts directed at understanding mechanistic controls.

### 3.4 Forced changes in phenology of net ecosystem production

Finally in this overview, we illustrate how anthropogenic forcing impacts the phase of the seasonal cycle by focusing on the phenology of NEP in the Northern Hemisphere middle to high latitudes (over 50°–80° N). NEP as a flux quantity represents the difference between gross primary production and ecosystem respiration, and thereby the net exchange of carbon with the atmosphere when fire and human land use changes are ignored. Our interest in NEP is motivated by eco-

logical concerns that a shift to an earlier spring bloom, in particular over the land regions adjacent to the Arctic, can drive a phenological mismatch in ecological interactions between plants and animals (Renner and Zohner, 2018). For the seasonality and phenology analysis in the upper panel of Fig. 5, an area integral of daily-mean NEP, is performed for each ensemble member separately. A total of 90 ensemble members are used, as daily-mean CLM5 output was not saved for the first 10 members, namely for members {1001.001, 1021.002, 1041.003, . . ., 1181.010}.

Ensemble-mean NEP is integrated over the region in 5-year intervals, with aggregation performed for individual years and with a binning interval of 1 day (colors in Fig. 5, upper panel). We find an evolving amplitude of the seasonal cycle and of the growing season length (the interval during which NEP is positive, indicating net land uptake of carbon). This representation of forced changes in the non-sinusoidal seasonal cycle reveals that the growing season length is projected to increase by almost 4 weeks, with the onset shifting 3 weeks earlier and termination shifting 1 week later. The forced changes in growing season length are mostly attributable to changes in the mean temperature (Lawrence et al., 2019; Lombardozzi et al., 2020). The analysis also reveals a more than doubling of the amplitude of the seasonal cycle in NEP as a forced response. This represents an increase in the "breathing" of the terrestrial high-latitude biosphere. Information from individual ensemble members in 20-year intervals regarding the timing of (i) first zero crossing, (ii) maximum NEP, (iii) second zero crossing, and (iv) maximum negative NEP (Fig. 5, lower panel) reveals that interannual variability (identified using one SD) is in general smaller than the forced trend evident in the ensemble mean in spring. Our analysis indicates that for the aggregated NEP signal, the phenological shift as a decadal trend already becomes emergent relative to the natural variability within the first decades of the 21st century. The trend itself is broadly consistent with observations (Zhu et al., 2016; Myers-Smith et al., 2020). Internal variability in the date of onset of the growing season decreases by 35 % over the course of the simulations and the date of the end of the growing season decreases by 18 % (Fig. 5, lower panel). In deriving these percentages, the transitions (zero crossings) were first calculated individually for each ensemble member for each time interval (across 90 members).

## 4 Summary and discussion

This study introduced a new, publicly available large ensemble of climate change simulations conducted with the global fully coupled CESM2 model. This large ensemble (CESM2-LE) is unprecedented in terms of its combination of size (100 members), duration (1850–2100), and spatial resolution in the atmosphere and ocean (nominally 1° horizontally). As such, it offers a unique opportunity to study not only forced changes in the mean state, but also forced changes in internal variability, including higher-order statistical moments. Here we showcase aspects of the remarkable diversity of forced responses in amplitude, frequency, patterns, co-variance, and seasonal characteristics of internal variability in CESM2-LE across a broad suite of key physical and ecosystem quantities, spanning the atmosphere, land, cryosphere, and ocean. Importantly, and contrary to conventional wisdom, the changes are not solely centered on the frequency of specific climate modes such as ENSO and the Madden–Julian oscillation CE3 but are instead broadly distributed over nearly all timescales (Fig. 2), in particular for non-Gaussian distributed variables. The mechanistic underpinnings of the changes in variability go beyond amplification or damping of major climate modes, and possibly include state dependence of linear stabilities, non-linearities, rectification, and changes in damping timescales and noise characteristics, many of which will be investigated in forthcoming studies analyzing the breadth of the CESM2-LE output fields.

If the ubiquitous changes in variance across temporal and spatial scales described here are realized in the real world, they will have several important implications for informing adaptation strategies and assessing potential impacts. This holds for water resource management and agriculture, fisheries, and occurrence of wildfires. Forced changes in phenology and phasing of the seasonal cycle for ecosystem productivity pose risks of mismatches with trophic-level interactions and energy transfers. The ubiquity of such changes in variability also points to the importance of moving beyond the assumption of stationary variability in detection and attribution studies of climate change (Hegerl et al., 2007) and underscores the necessity of recalibrating climate-economy models (Diaz and Moore, 2017) to account for an entirely different probability distribution for variability (Figs. 2 and S7) than what is currently used when projecting future climate change scenarios. The non-stationary nature of climate noise under anthropogenic forcing (Fig. 2) and the evolving teleconnections patterns (Fig. 4) also have implications for seasonal to multi-year climate predictability.

Although our analysis of CESM2-LE has revealed a broad range of forced changes in variance across physical scales and Earth system variables, it nevertheless should be emphasized that model uncertainty has not been considered here. There is already evidence for the narrower case of interannual variability in surface temperature and precipitation that model uncertainty in forced changes exhibits pronounced differences between models (Maher et al., 2021) (their Figs. 7 and 8 in the Supplement). Thus, it is our hope that our work will motivate further investigations of forced change in Earth system variance across a broad range of timescales under existing archives of large ensemble simulations (Deser et al., 2020; Schlunegger et al., 2020).

Taken together, we have provided support with new examples and new global emphasis that the Earth system is sensitive in its statistical characteristics to anthropogenic forcing,

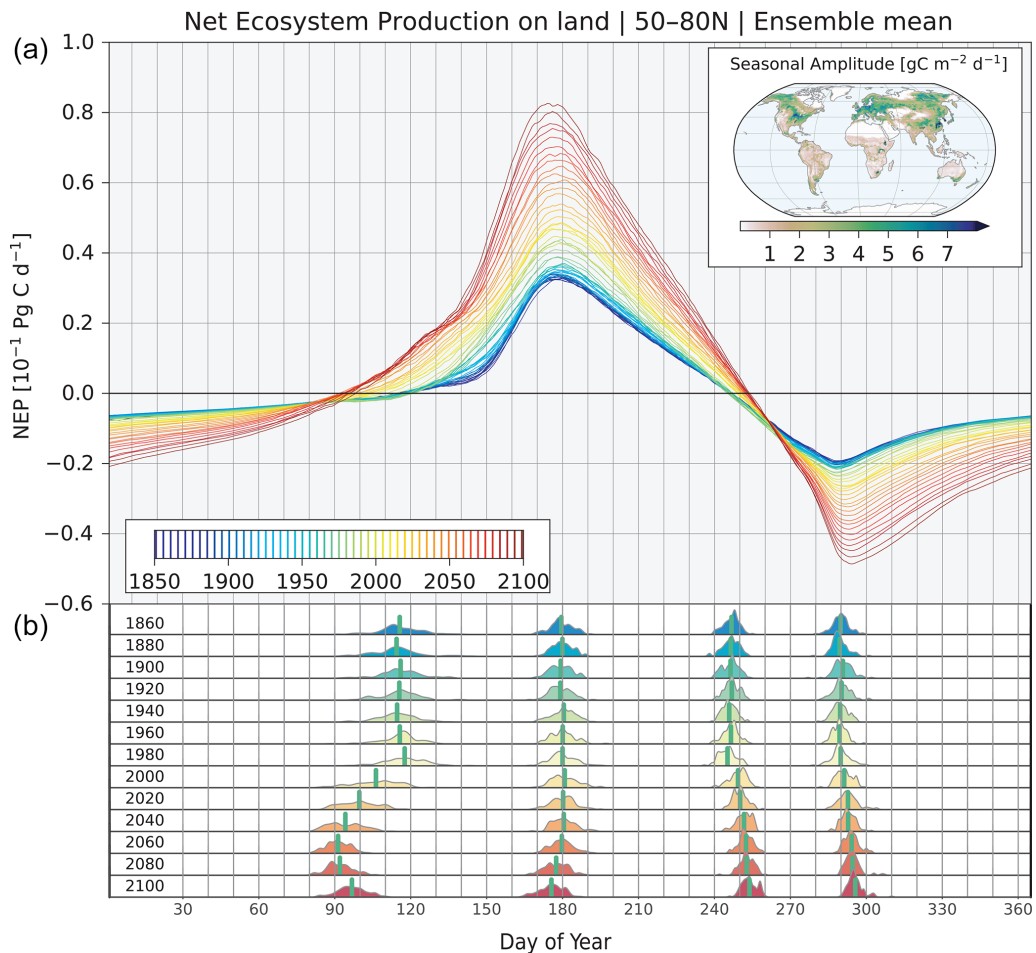

**Figure 5.** Expansion of growing season length, or equivalently the carbon uptake period, over 50°–80° N (shown here for all 90 members for which daily-mean land output was saved). **(a)** Evolution of ensemble-mean seasonal cycle (one line for every five years, color-coded) of integrated net ecosystem productivity (NEP), with positive values indicating net terrestrial carbon uptake and negative values indicating loss of carbon from the aggregated land region. The first zero crossing arks the start of the growing seasons, and the second zero crossing marks the end of the growing seasons. **(b)** Histograms of first occurrence of zero crossing, peak, second zero crossing, and minimum as a function of the day of the year. The horizonal axis for both panels is a climatological calendar day of the year, and aggregation is done across 90 members. The histograms represent model output sampled at 20-year intervals. The inlay map (upper right) shows the ensemble mean amplitude of the seasonal cycle of NEP averaged over 1960–1989 ($gC m^{-2} d^{-1}$).

thereby building upon and complementing previous studies that have focused on mechanistic analyses for specific phenomena (Swain et al., 2018; Tamarin-Brodsky et al., 2020; Taschetto et al., 2020; Burger et al., 2020). Although only a small fraction of such forced changes could be documented in this study, we expect that the diagnostic ensemble analysis tools applied here, along with the open access to our datasets, will inspire further investigations into the non-stationarity of Earth system processes in the presence of anthropogenic forcing.

**Code availability.** The code used to generate the figures in this study is available here: https://github.com/kj-stein/CESM2-LE (FIG_CODE_GENERAL, 2021).

The Python wavelet software used for Fig. 3 was provided by Evgeniya Predybaylo (Torrence and Compo, 1998) and is available at http://atoc.colorado.edu/research/wavelets/ (FIG_CODE_WAVELET, 2021).

**Data availability.** The CESM2-LE model output is available through https://www.cesm.ucar.edu/projects/community-projects/ LENS2/data-sets.html (CESM2_LE_OUTPUT, 2021).

**Supplement.** The supplement related to this article is available online at: https://doi.org/10.5194/esd-12-1-2021-supplement. TS21

**Author contributions.** The CESM2-LE project was initiated by KBR, AT, GD, and CD. The scientific framing of this paper was developed by KBR, AT, JEK, RY, KS, SSL, and MFS. Analyses and scientific post-processing were performed by RY, JEK, KS, LH, TB, and WK. The CESM2-LE model runs were set up, performed, and extracted through a joint effort by the team of SSL, NR, and JE. The initialization procedure for the model was developed through the joint efforts of CD, GD, IS, WK, SGY, and NR. All authors discussed the results and contributed to the writing of the paper.

**Competing interests.** The contact author has declared that neither they nor their co-authors have any competing interests.

**Disclaimer.** Publisher's note: Copernicus Publications remains neutral with regard to jurisdictional claims in published maps and institutional affiliations.

**Acknowledgements.** The authors would like to thank Alexis Tantet and an anonymous reviewer for their thoughtful and constructive comments.

The CESM2 large ensemble (CESM2-LE) simulations presented here for the first time have been conducted through a partnership between the IBS Center for Climate Physics (ICCP) in South Korea and the Community Earth System Model (CESM) group at the National Center for Atmospheric Research (NCAR) in the US, representing a broad collaborative effort between scientists from both centers.

The authors would like to thank Woncheol Roh at the ICCP and John Fasullo, Keith Lindsay, Adam S. Phillips, and Gary Strand at NCAR for their input and support. We would also like to thank all of the other scientists, software engineers, and administrators at both NCAR and the ICCP that contributed to this project. The framework for the macro- and micro-perturbation initialization strategy employed here also benefited from the US CLIVAR Workshop on large ensembles held in July 2019 in Boulder CO, USA, for which we wish to acknowledge the support of Mike Patterson, Jennie Zhu, and Jeff Becker at US CLIVAR.

The CESM project is supported primarily by the US National Science Foundation (NSF). This material is based upon work supported by the NCAR, which is a major facility sponsored by the US NSF under cooperative agreement 1852977. The CESM2 preindustrial control run was performed on the Cheyenne supercomputer (https://doi.org/10.5065/D6RX99HX) operated by the Computational and Information Systems Laboratory (CISL) at NCAR.

The simulations presented here with CESM2-LE were conducted on the IBS/ICCP supercomputer "Aleph", a 1.43 petaflop high-performance Cray XC50-LC Skylake computing system with 18 720 processor cores, 9.59 PB of disk storage, and 43 PB of tape archive storage. The CESM2-LE project duration was 15 months, and it generated 5.3 PB of data and used approximately 80 million CPU hours of computing time.

**Financial support.** The work of Keith B. Rodgers, Sun-Seon Lee, Axel Timmermann, Ryohei Yamaguchi, Ji-Eun Kim, Karl Stein, and Lei Huang was supported by the Institute for Basic Sciences (IBS), Republic of Korea, under IBS-R028-D1. Tamas Bodai was supported by the Institute for Basic Sciences (IBS), Republic of Korea, under IBS-R028-Y1. Malte F. Stuecker was supported by the NOAA Climate Program Office Modeling Analysis, Predictions, and Projections (MAPP) program, grant NA20OAR4310445, and participates in the MAPP Marine Ecosystem Task Force. This is IPRC publication 1540 and SOEST contribution 11412. William R. Wieder and Danica L. Lombardozzi were supported by the National Institute of Food and Agriculture, US Department of Agriculture (2015-67003-23485). William R. Wieder was also supported by NASA Interdisciplinary Science Program award NNX17AK19G. The work of Nan Rosenbloom was supported by the Regional and Global Model Analysis (RGMA) component of the Earth and Environmental System Modeling Program of the US Department of Energy's Office of Biological and Environmental Research (BER) via the US NSF IA 1947282. Eui-Seok Chung was supported by the project PE21010 of the Korea Polar Research Institute. NCAR is a major facility sponsored by the US NSF under cooperative agreement 1852977.

**Review statement.** This paper was edited by Valerio Lucarini and reviewed by Alexis Tantet and one anonymous referee.

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

## Remarks from the language copy-editor

## Remarks from the typesetter