# Peer review of "Ubiquity of human-induced changes in climate variability"

_Earth System Dynamics, 2021_

## Referee Comment (RC2)

**Review of « Ubiquity of human-induced changes in climate variability »**

The authors present a new dataset consisting in a large-ensemble simulation from an Earth system model with a relatively fine atmospheric resolution over a relatively long period (1850-2100) and representing a significant computational effort. They use this product to rather systematically describe changes in climate variability in the model, moving beyond analyses of climate change in terms of means. Variability is considered in terms of probability, distribution, amplitude, frequency, phasing, and patterns. Moreover, they study a large spectrum of patterns of climate variability, including the continuous part of the spectrum, rather than solely focusing on leading modes of climate variability such as El Nino-Southern Oscillation. The dataset and the systematic study of changes in climate variability on a broad range of time scales makes this study original. Moreover, its main conclusion that "the ubiquity of such changes in variability also points to the importance of moving beyond the assumption of stationary variability in detection and attribution studies of climate change" is timely.

This is why I recommend this article to ESD once the following minor comments have been addressed.

**Minor comments :**

**Abstract :**

l. 19 To emphasize the originality of the dataset, I would also mention the simulation period and the model resolution like at l. 333

l.23 « Greenhouse warming will in particular alter... ». I would rather write « Greenhouse warming in the model in particular alters... » and mention that model uncertainty is not considered here, as explained in the paragraph of l. 356.

**Introduction :**

l. 33 « spectral variance peaks ». The « peaks » are also relatively broad for different reasons contrary to spikes (Dirac's). I would write « relatively sharp peaks ».

l. 53 « variance » is repeated

l. 59 Explain why the the results of the CMIP6 version of CESM2 is expected to be conclusive compared to "earlier studies" (l. 46). Perhaps based on what is explained in Section 2.

**Method :**

l. 78 Missing punctuation

l. 81 Is the resolution for the POP ocean model also 1 degree ? It is important to know if it is eddy-resolving since this might also affect the atmospheric variability.

L. 91 – paragraph : How are these improvements measured ? To which extent does the land model (fire model and agricultural management in particular) rely on assumptions regarding human behavior in the future (for instance in response to climate change) ?

**Results :**

l. 169 Could you explain what motivated the choice of these observables ? I guess one factor is the relationship between these observables and climate-change impacts, but this is not obvious.

l. 169 Instead of the Fourier transform of the observable, why not use an estimate (e.g. periodogram) of the power spectrum which can be directly related to the variance that you use in Figue 1 (as the integral of an adequately normalized power spectrum) ? The variance is also used in Figure 3. If this is in fact what you are doing, please make it clearer.

l. 169 To avoid spectral leakage, a window should be applied before the FFT. Is this the case ?

l. 171 and l. 172 Why 35 years and not 30 years (2070-2099 and 1960-1989) ?

l. 173 If the power spectrum is computed, an alternative would be to first compute correlation functions for each member, average over the members and then do the FFT. I do not know which estimation method has the best properties, but could you explain why you made this choice ?

Figure 2 What are the units of the spectral amplitudes given the observable ?

l. 183 Even if I do not think that it is necessary to add confidence intervals to all panels and for all frequencies or to test the significance of the differences between spectra, could you give an estimate of what would be the width of these confidence intervals given the data that you use (in the supplementary material for instance) ? This would also make this section more coherent with the part on wavelets.

l. 200 and l. 208 Although a scalar observable can technically be seen as a bilinear form, I would reserve the term positive definite for non-trivial bilinear forms (e.g. represented by non-scalar matrices) and simply write "positive variables".

l. 202 This is not true for all stochastic processes. I guess you mean for a Brownian motion ?

l. 251 Could you clarify what is meant by "cross-ensemble" everywhere this expression is used ?

l. 256 "minimum n" → "minimum in"

l. 265 Same as for l. 251. "cross-ensemble calculations applied for identical time records for each ensemble member" is not clear to me. The standard deviation is computed from a sample combining all members and all years for a given period (historic or future) ? Based on the caption, I guess not, the standard deviation is computed over the ensemble and then time averaged over the period. Could you clarify and explain why you made this choice and not the other ?

l. 310 "and" → "an" I guess

Figure 5 : Do the histograms aggregate ensemble members for a single year only or is their also an aggregation of the 20 years in the interval (in which case the histograms would include the 20y-trend) ?

Figure 5 : How are the histograms estimated ? Using a grid ? Which interval length(s) ?

l. 318 Since the spacing between to vertical grid lines in Figure 5 represent an interval of 10 days, it seems to me that the shift in the onset is closer to 3 weeks or even 4 weeks than 2 weeks. Am I wrong ?

l. 322 How do you "measure" the interannual variability ? Even if you read the histograms with the eyes, I guess that you have some metric representation of the spread in mind, such as the standard deviation. In fact, if you use the distance between the minima and the maxima, the interannual variability appears comparable to the trend to me.

l. 325 Same question as for l. 322. Which measure do you use to obtain these percentages ?

**Discussion :**

l. 331 I would call this section "Summary and Discussion", but that's a detail.

l. 335 English is not my mother tong, so I may be wrong, but shouldn't "affords" be replaced by "offers" ?

References are hard to read because entries are not visually separated.

---

## Author Response (AR1)

RESPONSE TO FIRST REVIEWER

We thank the reviewer for their careful reading of the manuscript and constructive suggestions for improvements. The reviewer has thoughtfully suggested a valuable list of previous studies that we can reference that will broaden the context of our presentation, and to the best of our ability we have incorporated these references into our revisions. Additionally, the reviewer has highlighted several instances where we can comment on mechanistic underpinnings of model behavior, and in such cases we have also included appropriate references and commentary in the revised text.

Our detailed responses are below. For purposes of clarity, we have put the reviewer comments/questions in italicized text, and responded in plain text.

SPECIFIC COMMENTS

*ll. 33-37: I find a bit limiting the notion of fluctuations as "characterized by spectral variance peaks superimposed upon a broad noise background", as I think it does not entail the possibility that modes of spatio-temporal variability are in fact influenced by the "noise background" itself. Especially when one deals with processes that have clearly non-Gaussian PDFs, as in the case of this analysis, it is worth mentioning that at least multiplicative noise processes (and externally-driven changes therein) can alter the modes of variability through nonlinear interaction (e.g. Majda et al, 2009; Sardeshmukh and Sura, 2009; Sardeshmukh and Penland, 2015).*

We thank the reviewer for raising this point. Indeed the role of multiplicative noise in affecting variability on a multitude of timescales is well established in the literature. The references suggested by the reviewer are included in the revised draft, in addition to references to Mueller (1987; J. Phys. Oceanogr.), Levine and Jin (2010; JAS) and Jin et al. (2020, Simple ENSO Models in AGU monograph on El Niño in a changing climate). To our knowledge the Mueller paper is the first that describes the influence of multiplicative noise on second- and third-order cumulants and spectra in the context of the stochastic climate model. The other papers touch on or highlight the role of multiplicative noise in generating ENSO characteristics.

We will further revise the text by adding the following sentence: "The spectrum of observed regional-to-global climate fluctuations exhibits relatively sharp peaks and a broad noise background (Hasselmann, 1976; Franzke et al., 2020). Spectral peaks can emerge from a range of mechanisms, including astronomical forcings and internal climate instabilities, such as for ENSO. Moreover, these distinct features can be further influenced by climate processes acting on different timescales. Examples for this type of nonlinear "timescale interaction" are multiplicative (state-dependent) noise (Mueller, 1987; Majda et al., 2009; Sardeshmukh and Sura, 2009; Sardeshmukh and Penland, 2015; Jin et al., 2007; Levine et al., 2010; An et al., 2020; Jin et al., 2020) and combination mode dynamics (Stuecker et al., 2015).

*ll. 62-64: while I find that mentioning Milinski et al. (2020) objective algorithm for the detection of the required LE size is appropriate, I think that with the algorithm being model-dependent, it should be acknowledge that their conclusions do not a priori apply here. Possibly a sampling over the pre-industrial simulation, using it to test the internal variability*

*associated with ENSO, would hint at the number of members that is actually required (even though one would have to assume that the same holds when the SSP3-7.0 forcing is applied).*

The point of the reviewer regarding the model-dependence of the Milinski et al. (2020) study is well-taken, and this question is the subject of an independent study with CESM2 led by one of our coauthors, Tamas Bodai. It is indeed not possible to project the necessary ensemble size in a model based on findings from another model configuration, and there are other issues that complicate the idea of prescribing a general rule for ensemble sizes required for an experiment. As for the generic idea of an ensemble-size dependence of detectability and the accuracy of identifying forced changes, the following figure considers variance changes for Niño3 SST following the suggestion of the reviewer:

[Figure]

In the figure, rho denotes the detection rate, calculated with a bootstrap mean over 1e3 bootstrap samples, and F refers to the F-test for the slope, concerning whether it is significantly non-zero at the 95% significance level. HAC refers to a technique that considers heterscedasticity and auto-correlation – which are themselves in fact not detectable here, hence the agreement between $\rho_F$ and $\rho_{F,HAC}$. Thus the 20[th] century changes in the model are already detectable with 20 ensemble members.

The study of Milinski (2020) is not concerned with detectability, but rather with the accuracy of large changes, and this is addressed by the yellow and purple lines in the figure. $\alpha$ is the temporal slope of the ensemble standard deviation, and the purple line indicates the best

estimate from the 100 ensemble members.  $q_{97.5}$ and α are the 97.5$^{th}$ quantile (corresponding to the upper (u) bound of the 95% confidence interval) and the standard deviation over the bootstrap samples, respectively, with the relative errors/"variance" plotted.

In the revised text we have included clarification of this point as follows:

"The choice to use 100 ensemble members was motivated by the challenges associated with identifying trends in higher-order statistical moments. A previous set of analysis performed with the Max Planck Institute Grand Ensemble (MPI-GE) (Milinski et al., 2020) explored the relationship between ensemble size and the accuracy of identifying forced changes in higher-order moments. Even taking into account differences in model architecture and thereby model uncertainty in such estimates, their analysis with the MPI-GE nevertheless supports our decision to expand well beyond the 40 members chosen for the CESM1 LE (Kay et al, 2015).

*ll. 106-107:  the choice of the section of the pre-industrial run, where the model drift is particularly small, should be better justified. The internal variability of the model might be influenced by the presence (or absence) of such bias, and it would be relevant to assess how relevant this impact is.*

In the revised manuscript, we will provide a clearer description of model drift over the span of the initialization dates from the pre-industrial control run. In the CESM2 presentation paper of Danabasoglu et al. (2020), Fig 6 showed the TOA energy imbalance for the pre-industrial run over years 1-1200, revealing minimal drift by year 1000. Additional extensive diagnostics by NCAR scientists identified minimal drift for the AMOC, and for upper ocean heat content over the North Atlantic and Southern Ocean over years 1000-1300.

We have revised the manuscript text so that it now reads:

"…this corresponds to a time when model drift is relatively small, as is indicated  in the relatively small and stable top-of-the-atmosphere (TOA) global energy balance by year 1000 of the pre-industrial control run with CESM2 shown in Fig. 6a of Danabasoglu et al. (2020)."

*ll. 108-116:  I am a bit puzzled by the choice of the initialization dates for the ensemble.  80 members are initialized with 4 initial dates (sampled according to the phase of the AMOC; maximum AMOC, minimum AMOC, ascending AMOC, descending AMOC), then slightly perturbing these initial conditions (20 members per date); for the additional 20 members, initial dates separated by 10 years were chosen. I find hardly justifiable that the members are to be considered as independent, and identically distributed, and that as such, conclusions can be drawn about ensemble mean moments of the distribution. I acknowledge that, as the authors state at ll. 122-123, "further quantitative exploration of the specific duration over which initial condition memory is retained is the subject of a separate ongoing study", but I see two issues in this choice of the initial dates: 1.  Members chosen according to AMOC phase are not uncorrelated by construction; 2. When it comes to the internal variability of the ocean, it is quite unlikely that 10 years are a sufficient decorrelation time.*

We were not sufficiently clear in our submitted manuscript about issues surrounding the initialization strategy, through a mix of macro-and micro-perturbations. It was not our intention to argue that the initialization procedure for CESM2-LE produces members that can be considered as independent, and we should have stated this more clearly, we apologize for

the misunderstanding. Our analysis with internal variability is primarily focused on the post-1960 period, so more than a century after the 1850 initialization time for all members. In order to clear up any potential confusion, in addition to more detailed text clarifying the initialization procedure, we will provide in the revised manuscript both a quantification of the autocorrelation timescale for the AMOC, as well as a timeseries figure showing the evolution of AMOC transport over the late 19th century for the 4 sets of micro-perturbation runs, illustrating the timescale over which initial condition memory is lost.

[Figure]

This two-panel figure shows the evolution of AMOC for (top panel) individual ensemble members over 1850-2000, and (bottom panel) ensemble means for the four micro-perturbation groupings as well as the ensemble mean of the macro-perturbations, all considered at 26.5°N. By the year 1900, the bottom panel indicates convergence in the ensemble mean groupings. The bottom figure is now included in the Supplementary Materials.

[Figure]

This is consistent with the autocorrelation for the (detrended) AMOC calculated using years 401-2000 from the pre-industrial control run (piControl), as shown for both 26°N and 45°N in the above figure. This is consistent with our interpretation that the analyses in Fig. 2 and Fig. 4 occur well beyond the time when initial condition memory is important.

In the revised text we now say:

"It warrants mention that when using the CESM2 LE, the initialization procedure shot not be considered to produce members that are independent or to have randomized modes of climate variability for the years immediately subsequent to 1850. Taking as an example the AMOC strength at 26.5°N) (fig. S2a), the ensemble mean AMOC strength for each of the micro-perturbation clusters initialized for years 1231, 1251, 1281, and 1301 of the pre-industrial control run (averaged across 20 members for each case) converge only after several decades, indicative of the timescale over which initial condition memory persists for AMOC. This finds further support in a separate analysis of the autocorrelation timescale for AMOC variability at both 26.5°N and 45°N for the pre-industrial control run revealing that memory of initial conditions for AMOC do not extend beyond several decades. For this reason, our analysis with internal variability focuses on the period after 1960, more than a full century after initialization."

*ll. 130-136: I do not think that enough evidence is here provided that the two ensembles with different biomass burning can be assumed as being (or not being) part of the same population. An assessment through statistical tests (e.g. Mann-Whitney?) would here support such an argument.*

We fully agree with the reviewer that during the period of biomass burning perturbations (1990-2020, effectively) the full suite of 100 members should not be assumed to be part of the same population, but rather considered as two sets of 50 members. This is what motivated our Supplementary Fig. 2 in the submitted draft. There are two new manuscripts under development led by ICCP scientists (coauthors on this study) that deal explicitly with the impacts of biomass burning on the climate state. For the surface temperature, sea ice, and precipitation the response is only significant over the 1990-2020 interval of the biomass burning perturbation itself. This is the reason why we chose the intervals 1960-1989 and 2070-2099 for our emphasis on changes in variance, as the first of these is prior to the biomass burning perturbation, and the second is 50+ years after the perturbation.

[Figure]

[Figure]

LENS2 - Macro/Micro Perturbation Design

- ❏ 80 members
- ❏ 4 discrete AMOC states
- ❏ 20 micro-perturbation members for each AMOC state

1231   1251   1281   1301

1850 PI-Control

**20 micro members**
1-10: CMIP6 BMB forcing
11-20: Smoothed CMIP6 BMB forcing

Aug 2021

Upon further reflection, we also recognize that we have not been sufficiently clear in the main text about which members are grouped with the CMIP6 and SMBB representation of biomass burning. To that end we have included as Supplementary two schematic figures that have already been prepared for our online description of the model runs:

https://www.cesm.ucar.edu/projects/community-projects/LENS2/

namely the two figures shown immediately above, in our Supplementary Materials to facilitate understanding of the model output organization. Additionally, we state more clearly in the revised text (adding Section 2.4 for Minor Bug Fixes for clarity):

"The code base for the SMBB simulations incorporates corrections to minor bugs that were present in the first 50 ensemble members. This pertains to the $SO_2$, $SO_4$, and gas phase semi-volatile secondary organic aerosol (SOAG) emission datasets.  For $SO_2$ and $SO_4$, the "shipping" and "agriculture+solvents+waste" components of forcing were inadvertently switched during the historical-to-projection transition, or more specifically in 2015. The bug with $SO_2$ partitioning does not impact the results considered here,  given that its components are summed up before use. On the other hand, the issue with $SO_4$ datasets can impact the model state evolution because the particles sizes and numbers differ for the $SO_4$ components. The SOAG emissions are calculated from several hydrocarbons, and they were not recalculated after an earlier bug correction in covering units of the lumped species for the biomass burning emissions. This issue was corrected, and diagnostics indicate that there are not any significant changes in the model solutions from these particular corrections with aerosols."

 As a related point, we will also state more explicitly that:

"The second correction introduced for the 50 BB_CMIP6_SM simulations concerns the presence of a sporadic large $CO_2$ uptake over land that occurred for the CMIP6 runs, with this associated with a negative flux of carbon occurring at crop harvest time over a single time step.  Although these large negative carbon flux component terms in autotrophic respiration are necessary for maintain carbon balance, the large $CO_2$ spikes are not realistic. To avoid

these spikes, the associated $CO_2$ fluxes that occur over a single time step are distributed to the atmosphere over a timescale of approximately six months for the SMBB simulations. Analysis indicates that these bug fixes for carbon between the CMIP6 and SMBB simulations did not result in any climate changing impacts."

*ll. 175: out of curiosity, I was wondering why the authors chose to take into account the maximum transport at 40°N, instead of 26.5°N (which is often considered as an AMOC metric).*

We agree with the reviewer that for consistency, the AMOC as represented in both Fig. 1 and Fig. S5 should be analyzed at 26.5°N. We have modified Fig. S5 accordingly for the revisions.

*ll. 201-202: as the authors refer here to variance and extremes, and their changes in future climate, it might be worth noticing that some promising results have been achieved with methods that synthesize several or all moments of the PDF, e.g. the Wasserstein distance (cfr. Ghil 2015; Robin et al., 2017; Vissio et al., 2020 for a climate model diagnostics application).*

We thank the reviewer for pointing to these earlier publications, we now reference them in the revised manuscript.

*l. 227: I do not have clear why the authors decided to retain the seasonal cycle in this context.*

We thank the reviewer for raising this point regarding the retention of the seasonal cycle in the wavelet analysis shown in Fig. 3, as we were not sufficiently clear about this in the submitted manuscript. Our reason for retaining the seasonal cycle stems from our interest in illustrating timescale interactions between ENSO and the seasonal cycle with the full power of large ensemble statistics. It is our hope that this will stimulate, as part of our presentation, further investigations of insights that are offered into frequency entrainment, among other questions that could arise. We now state explicitly in the revised text:

"Our motivation for retaining the seasonal cycle stems from an interest in illustrating timescale interactions between ENSO and the seasonal cycle with the full power of the Large Ensemble statistics. The annual cycle and ENSO interact with each other in a complicated way, with the annual cycle itself being a forced mode (Xie, 1994). This interaction gives rise to combination modes (Stuecker et al., 2015b), frequency entrainment (Timmermann et al., 2007), and ENSO's phase-locking and seasonal variance modulations (Stein et al., 2014; Stein et al., 2010). Not only does the annual cycle in the equatorial Pacific influence the amplitude and phase of ENSO, but ENSO also impacts the seasonal cycle."

*l. 246-247: this is one of a few sentences I found in the text, that justify my general comment above about the lack of interpretation. In particular, the authors mention a lead-lag relation between precipitation and SST seasonal maxima. The assessment of these relations are challenging in the context of climate models (Lembo et al., 2017), together with their interpretation (cfr. Su et al. 2005 for this this specific context) and the authors might want to discuss what these mean in terms of dynamics of the systems.*

We agree with the reviewer that there is an opportunity here to reference published literature that presents mechanistic interpretations of the behavior we have highlighted for the CESM2-LE. For the specific issue raised here of maximum precipitation leading maximum temperature over the Niño3.4 region on seasonal timescales (red dots in Fig. 3c and 3d), current scientific understanding maintains that precipitation is largely driven by meridional SST gradients, and is thereby not directly tied in its phasing to local SST. In other words, moisture convergence is in part determined by non-local SST conditions. We now appropriately reference the studies of Xie (1996), Xie et al. (2010), Williams and Patricola (2018), and Stuecker et al. (2020) on this topic in the revised text.

*l. 276*: *This is in part already known. Several studies (e.g. Screen 2014, Chen et al. 2016, Haugen et al., 2018) have shown evidence of a relationship between Arctic amplification and reduced temperature variance over the mid- and high-latitudes of the Northern Hemisphere, and an interpretation of this has been given from a dynamical point of view (cfr. Sun et al., 2015; Schneider et al., 2015), involving the role of precipitation.*

We thank the reviewer for suggesting that appropriate references be given for describing changes in variance in temperature over the mid-to-high latitudes of the Northern Hemisphere due to polar amplification. We now reference these studies in the revised version of the manuscript, as well as the studies of Holmes et al. (2016; J. Climate) and Screen et al. (2015; BAMS).

*ll. 322-323: same as in my comment to l. 276. I am not surprised that the authors find a reduction in the NEP inter-annual variability, as this is linked to the variability of near-surface temperature. The link has been discussed in previous works (e.g. Yao et al., 2021), and I believe it should be taken into account here.*

It seems that the reviewer may in fact be confused by the text in our manuscript. We specifically chose not to address interannual variability in NEP. Rather our focus was on interannual variability in phenology in the lower panel of Fig. 5, as this is the behavior that has to our knowledge not been previously described in published literature. In our revisions, we will move the interpretation of the cause of the forced trend earlier in the paragraph, where we discuss the time of emergence of the trend.

In order to clarify, we have modified text at the end of our discussion of phenology to say: "Our analysis indicates that for NEP aggregated over this region the phenological shift as a decadal trend becomes emergent relative to estimates of the natural variability already within the first decades of the 21st century, a trend that is broadly consistent with observations (Zhu et al., 2016; Myers-Smith et al., 2020). The forced changes in growing season length are mostly attributable to changes in the mean temperature (Lawrence et al., 2019; Lombardozzi et al., 2020). Internal variability in the date of the onset of the growing season decreases by 35% over the course of the simulations and decreases by 18% for the date of the end of the growing season (Fig. 5, lower panel)."

The statements pertaining to attribution (expansion of growing season length being attributed to temperature) have been moved to the beginning of the same paragraph, so that the attribution is more of an emphasis in the text.

*ll. 328-330: see my comment at ll. 246-247. I think the authors should comment on this finding and on how this can be interpreted.*

The paragraph pointed to is introducing the climate change impacts on the mean state and variability. The reviewer is asking us to interpret the result related to changes in variability of peak and trough NEP amplitude (mentioned in liens 328-330, changes in variability along the y-axis of Fig. 5), but we don't believe that this is where the main story lies here. Instead, we focus on the discussion of the temporal trends and changes in the seasonal variability related to spring-green up (x-axis of Fig. 5). Specifically, we focus on changes in the mean state related to forced changes in phenology, notably the earlier initiation of the growing season in the Northern Hemisphere spring, where ecological functions may be disrupted. Under future scenarios we also see reduced variability in the first zero-crossing of NEP, which we attribute to the combined effects of warming and the timing of snowmelt. A subset of the coauthors of this manuscript are pursuing this question of drivers of phenology as an offshoot project that will complement the presentation paper analysis.

*ll. 350-351: this is not a new achievement, it has been long known (see, e.g. Palmer 1993; Corti et al., 1999) that climate change projects on modes of variability in several ways.*

The point raised here by the reviewer has been challenged by published literature (Ghil and Lucarini, Rev. Mod. Phys., 92, 2020; Gritsun and Lucarini, Physica D 349, 62, 2017). As a detailed discussion of this point would serve as a distraction from our study, we leave this as a possible topic for future investigation.

*l. 360: I wonder if the authors are able to comment on how significant these findings obtained with CESM2-LE are, in relation with other Large Ensemble exercises described in Maher et al., 2021.*

The reviewer refers to the multi-model Large Ensemble intercomparison study of Maher et al. (2021), and by implication the growing number of studies that make use of the multi-model Large Ensemble Archive presented in the study of Deser et al. (2020). From the onset of our project with the CESM2-LE, our intention has been to have our simulations be available for such multi-model studies, and to that end we specifically chose the historical/SSP3-7.0 pathway recommended in the CMIP6 protocols (ScenarioMIP) study of O'Neill et al. (2016). We have opted here to not engage in an intercomparison exercise, as this is beyond the scope of this initial presentation of the CESM2-LE itself, but to reiterate we have made every effort to facilitate such inter-comparison studies by interested parties in the future. Nevertheless we wish to note that a complementary effort has developed to the point of a submitted manuscript including comparisons across Large Ensemble models (Santer et al., "Robust anthropogenic signal identified in the seasonal cycle of tropospheric temperature"), with other such inter-comparison manuscripts expected in the near-term.

We have also made available (https://climatedata.ibs.re.kr/data/cesm2-lens/lens-diagnostics) the results of the Climate Variability Diagnostics Package for Large Ensembles (CVDP-LE) of Philips et al. (2020) (https://www.cesm.ucar.edu/working_groups/CVC/cvdp-le/) for diagnostics over a broad suite of variables from our simulations, as a means to facilitate studies that seek to understand model differences. We will include a link to this with an explanation in the revised version of the manuscript.

*l. 364-367: the lack of interpretation of the findings is evident here. I don't think that the take-home message is that the Earth system is "far more sensitive in its statistical characteristics to anthropogenic forcing than previously recognized". There is actually a*

*literature on assessing changes in higher order moments of several aspects of climate variability, often using Large Ensemble exercises, e.g. Swain et al., 2018, for regional precipitation, Tamarin-Brodsky et al., 2020, for NH temperature variability, among others. The authors might compare their results with others, in order to explain how the sensitivity of statistical characteristics was less recognized before. As mentioned above, some of the findings, taken one by one, are confirming, or possibly expanding, what was already somehow known from previous works. The manuscript might be significantly improved, if the authors would at least qualitatively discuss what drives, and what is the relation between e.g. changes in frequency and phasing of ENSO w.r.t. SST and precipitation, cross-ensemble SD for temperature and precipitation, change in ENSO's remote correlation with regional mean temperatures and precipitation over some regions, just to mention a few features that might be interpreted in the light of changes occurring in the general circulation.*

Yes, we agree with the reviewer that the sentence towards the conclusions of the manuscript that "the Earth system is far more sensitive in its statistical characteristics to anthropogenic forcing than previously recognized" would be better rephrased as "we have provided support with new examples and new global emphasis that the Earth system is sensitive in its statistical characteristics to anthropogenic forcing, thereby building on previous studies". With regard to the questions posed by the reviewer with regard to Fig. 4, namely ENSO teleconnections, we will also reference properly the AGU monograph published in 2020 entitled "El Niño Southern Oscillation in a Changing Climate", in particular the chapter by Taschetto et al. entitled "ENSO Atmospheric Teleconnections". This synthesis reference appropriately addresses the challenges for understanding how ENSO teleconnections can change, including the relative role of local diabatic forcing and modulations of ENSO for understanding regional responses.

TECHNICAL CORRECTIONS

*l. 109: replace "runs" with "run"*

We have corrected this error in the revised manuscript.

*l. 150: replace "weaking" with "weakening"*

This has also been corrected, following the suggestion of the reviewer.

*l. 209: replace "associate" with "association"*

This has also been corrected.

*l. 256: replace "n" with "in"*

This has also been corrected.

RESPONSE TO SECOND REVIEWER

We would like to thank the reviewer for offering a number of insightful and constructive comments and criticisms, which we address below. In the text that follows, the reviewer's comments are indicated in italicized text, and our responses are shown in plain text.

Minor Comments:

Abstract:

*l. 19: To emphasize the originality of the dataset, I would also mention the simulation period and the model resolution like at l. 333.*

We thank the reviewer for the suggestion, this has also been clarified in the revised manuscript. We state explicitly that the runs were performed over 1850-2100.

*l. 23: "Greenhouse warming will in particular alter…". I would rather write "Greenhouse warming in the model in particular alters…" and mention that model uncertainty is not considered here, as explained in the paragraph of line 356.*

For this presentation paper of the CESM2-LE, model uncertainty is neither a conclusion or our study or a scientific implication of our study. For that reason we have elected to not mention model uncertainty in the Abstract, but rather it is mentioned in the next-tofinal paragraph of the study.

Introduction:

*l. 33: "spectral variance peaks". The "peaks" are also relatively broad for different reasons contrary to spikes (Dirac's). I would write "relatively sharp peaks".*

We have made the change recommended by the reviewer.

*L. 53: "variance" is repeated.*

We have removed the redundant use of "variance".

*L. 59: Explain why the results of the CMIP6 version of CESM2 is expected to be conclusive relative to "earlier studies" (l. 46). Perhaps based on what is explained in Section 2.*

To address this point, we now state in the Introduction:

"A large number of improvements have occurred since the CESM1-LE (Kay et al., 2015), as documented in the Methods section…"

And then in the Methods section:

"CESM2 offers a number of improvements pertinent to our scientific intersts relative to what was available for the CESM1-LE (Kay et al. 2015). These improvements include advances

in the surface boundary layer representation for the ocean (Li et al, 2016), as well as for cloud microphysics (Gettelman et al., 2015).”

Method:

*l. 78: Missing punctuation*

The problem with the missing punctuation has been fixed.

*L. 81: Is the resolution for the POP ocean model also 1 degree? It is important to know if it is eddy-resolving since this might also effect the atmospheric variability.*

We now state explicitly in the main text:

“The nominal resolution of the ocean is 1° horizontally, with uniform spacing of 1.125° in the zonal direction and varying significantly in the meridional direction, with the finest resolution of ~0.25° at the equator.”

*l. 91 – paragraph: How are these improvements measured? To which extent does the land model (fire model and agricultural management in particular) rely on assumptions regarding human behavior in the future (for instance in response to climate change)?*

We thank the reviewer for pointing out that we were not sufficiently clear in the paragraph starting on line 91, in particular in describing the strengths of the land model and the benchmarking that has been done to assess skill in the model. We have reorganized the paragraph originally at line 91 to say:

“An important advance of great value to Large Ensemble investigations is achieved through new developments incorporated into the Community Land Model Version 5 (CLM5) (Danabasoglu et al., 2020; Lawrence et al., 2019; Lombardozzi et al., 2020). This model addresses a number of well-known limitations relative to previous versions of CLM, including enhanced simulated cumulative $CO_2$ uptake over the historical period (Bonan et al., 2019) improved representation of the seasonal cycle of net ecosystem production (NEP) (Lawrence et al, 2019), which is highlighted in our analysis of projected forced phenology changes. Improvements in CLM are found across a broad range of simulated variables that have been documented through evaluation of model simulations against the International Land Model Benchmarking (ILAMBv2.1) package and other analyses (Collier et al., 2018; Danabasoglu et al. 2020; Lawrence et al., 2019; Wieder et al., 2019). Notable features also included in CLM5 are the explicit representation of agricultural management and improvements in the implementation of the prognostic fire model (Lombardozzi et al, 2020; Lie et al., 2013; Li and Lawrence, 2017). We note that land model trajectories are sensitive to the SSP-RCP scenarios that determine the spatial distribution and extent of land use and land cover change (O'Neill et al., 2016). “

Results:

*l. 169: Could you explain what motivated the choice of these observables? I guess one factor is the relationship between these observables and climate-change impact, but this is not obvious.*

We thank the reviewer for indicating the need to clarify this point. To this end, we have added new text that states explicitly:

"The choice of these variables reflects our interest in both climate and ecosystem dynamics, as well as their societal relevance in terms of adaptation and resource management".

*l. 169: Instead of the Fourier transform of the observable, why not use an estimate (e.g. periodogram) of the power spectrum which can be directly related to the variance that you use in Figure 1 (as the integral of an adequately normalized power spectrum)? The variance is also used in Figure 3. If this is in fact what you are doing, please make it clearer.*

In the caption for Fig. 2 and in l.169 we specified "amplitude spectrum", which is the square root of the power spectrum. Please note that Fig. 1, Fig. 3e and Fig. 3f are cross-ensemble standard deviations using annual mean data (the annual mean is calculated first for each member, and then the standard deviation is taken across ensemble members for the same year). Furthermore, they are not representing "variance". Thus, even if we use power spectral density (PDS) in Fig. 2, the integral of PSD is not equivalent to what is shown in either Fig. 1 or Fig. 3. We chose the amplitude spectrum in Fig. 2 to have a better understanding of the amplitudes of perturbations at different timescales, and to be consistent with the standard deviations shown in the other figures. Also, if PDS were used, the already strong annual cycle and its harmonic peaks in the amplitude spectrum would become too large, making other spectral behaviors relatively less variable.

*l. 169: To avoid spectral leakage, a window should be applied before the FFT, is this the case?*

To address this point, we now state explicitly:

The point is well-taken, but the large degree of aggregation used for Fig. 2 (both across ensemble members but also spatial aggregation) alleviates the need for windowing. We will state this more clearly in our revised text.

*l. 171 and 172: Why 35 years and not 30 years (2070-2099 and 1960-1989)?*

We thank the reviewer for catching this typo, we have modified "35 years" to now say "30 years".

*l. 173: If the power spectrum is computed, an alternative would be to first compute correlation functions for each member, average over the members and then do the FFT. I do not know which estimation method has the best properties, but could you explain why you made this choice?*

There is a disparity between dynamical characteristics (such as power spectra) and statistical characteristics (like the natural measure) in that we can – for now – define only the latter in an instantaneous/snapshot or pullback sense. Given that there is no reference for the power spectrum with respect to which biases are defined it is not really possible to rank different possibly estimators. We went with what appears to be the most intuitive estimator: the E-mean of the temporally computed power spectra (or FFT amplitudes). A benefit of this

quantity is that nonergodicity could be naturally defined considering the difference between this and the correct, conceptually sound quantity.

*Figure 2: What are the units of the spectral amplitudes given the observables?*

Please find this in the caption for Fig. 2 in the original submitted manuscript, where we state "Spectra are shown as amplitude, with the units being the same as the x-axis for the PDFs."

*l. 183: Even if I do not think that it is necessary to add confidence intervals to all panels and for all frequencies or to test the significance of the differences between spectra, could you give an estimate of what would be the width of these confidence intervals given that data that you use (in the supplementary material for instance)? This would also make this section more coherent with the part on wavelets.*

[Figure]

We appreciate the reviewer raising this point of significance. To address this, we have added an additional figure (fig. S8) to our Supplementary Materials, illustrating an example of significance analysis for our spectral analysis.

"To demonstrate the significance of the spectra considered in Fig. 2 and fig. S7, an example is given in fig. S8 for precipitation over the Niño3.4 region for the same 1960-1989 (blue) and 2070-2099 (red) time intervals. The grey shading indicates the 95% confidence interval (1.95 x standard error). For each ensemble member, we first spatially averaged the spectra at individual grid points over the Niño3.4 region, and then calculated the standard error using

the 100 spectra for the full 100 ensemble members. This approach avoids sampling spatially correlated data in the calculation of the confidence intervals. The estimated confidence intervals in Fig. S8 indicate the spectra for the 1960-1989 and 2070-2099 are statistically different. It is worth noting here that if samples at different grid points are treated as being independent samples, the confidence intervals become much narrower, in which case the two spectra in fig. S8 are even more statistically significant in their differences"

*l. 200 and l. 208: Although a scalar observable can technically be seen as a bilinear form, I would reserve the term positive definite for non-trivial bilinear forms (e.g. represented by non-scalar matrices) and simply write "positive variables".*

We thank the reviewer for making this point, and we have switched to using the term "positive variables", as suggested.

*l. 202: This is not true for all stochastic processes. I guess you mean for a Brownian motion?*

We have modified the text from saying "stochastic processes" to say "white noise processes", we thank the reviewer for expressing the need to be clearer here.

*Could you clarify what is meant by "cross-ensemble" everywhere this expression is used?*

Yes, we will do this.

*l. 256: "minimum m" => "minimum in"*

This is correct, we have made the change as suggested.

*l. 265: same as for l. 251. "cross-ensemble calculations applied for identical time records for each ensemble member" is not clear to me. The standard deviation is computed from a sample combining all members and all years for a given period (historic or future)? Based on the caption, I guess not, the standard deviation is computed over the ensemble and then time-averaged over the period. Could you clarify and explain why you made this choice and not the other?*

We apologize for not being clearer, we have now clarified this in the text by stating:

"Cross-ensemble calculations here entail first calculating the standard deviation across all ensemble members for the same time record. Subsequently averaging is done across time. This sequence was chosen to avoid spurious amplification of variability due to the non-trivial forced variations in precipitation and surface temperature driven by volcanic aerosols over the historical period."

*l. 310: "and" => "an" I guess*

Yes, this is correct, we have modified the text accordingly.

*Figure 5: Do the histograms aggregate ensemble members for a single year only or is there also an aggregation of the 20 years in the interval (in which case the histograms would include the 20y-trend)?*

The aggregation is done for individual years, we have clarified this in the text.

*Figure 5: How are the histograms estimated? Using a grid? Which interval length(s)?*

The width of binning for the histograms is 1-day, we have clarified this in the revised text, we thank the reviewer for pointing out that this wasn't clear in the reviewed manuscript.

*l. 318: Since the spacing between the vertical grid lines in Fig. 5 represents an interval of 10 days, it seems to me that the shift in the onset is closer to 3 weeks or even 4 weeks than 2 weeks. Am I wrong?*

We thank the reviewer for raising this question, have replaced the misstatement of the duration of the onset shift to reflect that it is in fact three weeks. Comparing the mean over 1860-1869 and the mean over 2090-2099, the day of the first zero-crossing shifts 20.9 days and the day of the termination shifts 6.3 days.

*l. 322: How do you "measure" the interannual variability? Even if you read the histograms by eye, I guess that you have some metric representation of the spread in mind, such as the standard deviation. In fact, if you use the distance between the minima and the maxima, the interannual variability appears comparable to the trend to me.*

We have clarified this in the text, we used one standard deviation to measure the variability.

*l. 325: Same question as for l. 322: Which measure do you use to obtain these percentages?*

We thank the reviewer for raising this point, as we were insufficiently clear in the text. The calculations were performed year-by-year, where the transitions (zero crossing) were calculated for each member across the full set of 90 ensemble members as this progressed in time. This is now stated explicitly in the revised text.

Discussion:

*l. 331: I would call this section "Summary and Discussion", but that's a detail.*

We have followed the suggestion of the reviewer here, and changed the title of the section.

*l. 335: English is not my mother tongue, so I may be wrong, but shouldn't "affords" be replaced by "offers"?*

Yes, we have made the appropriate change so that the text says "offers".

*References are hard to read because entries are not visually separated.*

We have done our best to fix the formatting for the revised version of the manuscript to facilitate reading the references.

RESPONSE TO PUBLIC COMMENT

We appreciate the comments offered through the public comment on our manuscript. The core suggestion of the reviewer concerns our choice of scenario (SSP3-7.0) and our choice to run our simulations through the year 2100 so as to consider variance changes at the end of the 21$^{st}$ century, rather than choose the near-term years 2040 or 2050. Our decisions in these matters were anchored in community-based decisions as reflected for example in the O'Neill et al. (2016) ScenarioMIP paper, that suggested SSP3-7.0 for large ensemble simulations. And more broadly, we chose to follow in most ways the CMIP6 protocols that were developed through broad community decision-making over the last 5 years. We wish in no way to denigrate or dissuade research focusing on nearer-term changes, nor does our work endorse or "choose" most likely outcomes of political decisions or put our money on the most likely scenario for future change. The O'Neill et al. (2016) study was quite specific in its recommendation that as a relatively strong scenario, SSP3-7.0 offers relatively strong forcing, with this being appropriate for studying changes of variance over the 21$^{st}$ century. We're sorry for any misunderstandings in this regard. In the revised text, we will state more clearly how our model configuration was chosen within the context of broader CMIP6 efforts.

As a matter of procedure, we would encourage the reviewer to participate in the development of protocols for CMIP7, as this is where the protocols that shape studies such as ours are developed and expressed to the climate community. To reiterate, the interests and questions raised by the reviewer are clearly of value and interest for enhancing both public awareness and policy. But procedurally the most constructive way to bring such concerns to the table may not be through arguing posteriori that submitted manuscripts have illegitimate priorities for their chosen timescales (is any timescale illegitimate in climate science?), but rather in shaping community priorities through open processes.

REFERENCES CONSIDERED WHILE REVISING THE MANUSCRIPT

An, S.-I. et al. (2020), Fokker-Planck dynamics of the El Niño-Southern Oscillation, Scientific Reports 10(1), 1-11.

Burger, F.A., J.G. John, and T.L. Frölicher (2020), Increase in ocean acidity and extremes under increasing atmospheric $CO_2$, Biogeosciences, 17, 4633-4662.

Chen, H.W., et al. (2016), The Robustness of Midlatitude Weather Pattern Changes due to Arctic Sea Ice Loss, Journal of Climate, 29(21), 7831-7849.

Ghil, M., (2015), A Mathematical Theory of Climate Sensitivity, or How to Deal with Both Anthropogenic Forcing and Natural Variability? Book chapter.

Ghil, M., and Lucarini, V. (2020), The physics of climate variability and climate change, Rev. Mod. Phys., 92(3), 035002, 7.

Haugen, M.A, et al. (2018), Estimating changes in temperature distributions in a Large Ensemble of climate simulations using quantile regression, J. Clim., 31(20), 8573-8588.

Holmes, C.R., et al. (2016), Robust future changes in temperature variability under greenhouse gas forcing and the relationship with thermal advection, Journal of Climate, 29, 2221-2236.

Jin, FF, et al. (2007), Ensemble-mean dynamics of the ENSO recharge oscillator under state-dependent stochastic forcing, Geophysical Research Letters, 34 (3).

Jin, F.-F., et al. (2020), Simple ENSO Models. In El Niño Southern Oscillation in a Changing Climate (eds. M. J. McPhaden, A Santoso, and W. Cai), doi:10.1002/9781119548165.ch6.

Lembo, V., I. Bordi, and A. Speranza (2017), Annual and semiannual cycles of midlatitude near-surface temperature and tropospheric baroclinicity: reanalysis data and AOGCM simulations, Earth Syst. Dynam., 8, 295-312.

Levine, A.F.Z., and F. Jin (2010), Noise-induced instability in the ENSO Recharge Oscillator, Journal of Atmospheric Sciences, 67(2), 529-542.

Majda, A., C.L.E. Franzke, and D. Crommelin (2009), Normal forms for reduced stochastic climate models, Proceedings of the National Academy of Sciences of the United States of America, 106, 3649-3653.

Mueller, D. (1987), Bispectra of sea surface temperature anomalies, J. Phys. Oceanog., 17, 26-36.

Robin, Y., P. Yiou, and P. Nancon (2017), Detecting changes in forced climate attractors with Wassterstein distance, Nonlin. Processes Geophys., 24, 393-405.

Sardeshmukh, P.D., and P. Sura (2009), Reconciling non-Gaussian climate statistics with linear dynamics, Journal of Climate, 22(5), 1193-1207.

Sardeshmukh, P.D., and C. Penland (2015), Understanding the distinctively skewed and heavy tailed character of atmospheric and oceanic probability distributions, Chaos, 25(3), 036410.

Schneider et al. (2015), Physics of Changes in Synoptic Midlatitude Temperature Variabiltiy, Journal of Climate, 28(6).

Screen, J. (2014), Arctic amplification decreases temperature variance in northern mid-to-high-latitudes, Nature Clim. Change, 4, 577-582.

Screen, J.A., C. Deser, and L. Sun (2015), Reduced risk of North American cold extremes due to continued sea ice loss, Bull. Amer. Met. Soc., 96, 1489-1503, doi:10.1175/BAMS-D-14-00185.1.

Stein, K., et al. (2010), Seasonal synchronization of ENSO events in a linear stochastic model, Journal of Climate 23 (21), 5629-5643.

Stein et al., 2014, ENSO seasonal synchronization theory, Journal of Climate, 27, 5285-5310.

Su, H., Neelin, J.D., and Meyerson, J.E. (2005), Mechanisms for Lagged Atmospheric Response to ENSO SST Forcing, Journal of Climate, 18(20), 4195-4215.

Sun, L., C. Deser, and R.A. Tomas (2015), Mechanisms of stratospheric and tropospheric circulation response to projected Arctic sea ice loss. J. Climate, 28, 7824-7825, doi:10.1175/Jcli-d-15-0169.1.

Swain, D.L. et al. (2018), Increasing precipitation volatility in twenty-first century California, Nature Climate Change.

Tamarin-Brodsky, T., et al. (2020), Changes in Northern Hemisphere temperature variability shaped by regional warming patterns, Nature Geoscience.

Taschetto, A., et al. (2020), ENSO atmospheric teleconnections, in El Niño Southern Oscillation in a Changing Climate (eds. M.J. McPhaden, A. Santoso, and W. Cai) ), doi:10.1002/9781119548165

Timmermann, A., et al. (2007), The effect of orbital forcing on the mean climate and variability of the tropical Pacific, Journal of Climate, 20 (16), 4147-4159.

Vissio, G., et al. (2020), Evaluating the performance of climate models based on Wasswerstein distance, GRL.

Williams, I.N., and C.M. Patricola (2018), Diversity of ENSO events unified by convective threshold sea surface temperature: A nonlinear ENSO index, Geophysical Research Letters, 45, 9236-9244.

Xie, S.P. (1996), Effects of seasonal solar forcing on latitudinal asymmetry of the ITCZ, J. Climate, 9, 2945-2950.

Xie et al. (2010), Global warming pattern formation:  sea surface temperature and rainfall, Journal of Climate, 23, 966-986.